# Carbon degradation and mobilisation potentials of thawing permafrost peatlands in Northern Norway inferred from laboratory incubations

Sigrid Trier Kjær[1,2], Sebastian Westermann[2,3], Nora Nedkvitne[1], Peter Dörsch[1,3]

[1] Faculty of Environmental Sciences and Natural Resource Management, Norwegian University of Life Sciences, NMBU, Ås, 1433, Norway
[2] Department of Geosciences, University of Oslo, Oslo, 0371, Norway
[3] Centre for Biogeochemistry in the Anthropocene, University of Oslo, 0371, Norway

*Correspondence to:* Sigrid Trier Kjær (sigrid.trier.kjar@nmbu.no**)**

**Abstract.** Permafrost soils are undergoing rapid thawing due to climate change and global warming. Permafrost peatlands are especially vulnerable since they are located near the southern margin of the permafrost domain in the zones of discontinuous and sporadic permafrost. They store large quantities of carbon (C) which, upon thawing, may be decomposed and released as carbon dioxide ($CO_2$), methane ($CH_4$) and dissolved organic carbon (DOC). This study characterises patterns of potential C degradation and mobilisation within an area with sporadic permafrost by evaluating C degradation in three permafrost peatland ecosystems in Finnmark, Norway under laboratory conditions. Active layer, transition zone and permafrost samples from distinct cores were thawed under controlled conditions and incubated for up until 350 days under initially-oxic or anoxic conditions while measuring $CO_2$, $CH_4$ and DOC production. Carbon degradation varied among the three peat plateaus but showed a similar trend over depth with largest $CO_2$ production rates in the upper active layer and the top of the permafrost. Despite marked differences in peat chemistry between the layers, post-thaw $CO_2$ production of permafrost peat throughout the first 350 days reached 67-125% of that observed in samples from the top of the active layer. De novo $CH_4$ production occurred after prolonged anoxic incubation in samples from transition zone and permafrost, but not in active layer samples. $CH_4$ production was largest in incubations from thermokarst peat sampled next to decaying peat plateaus. DOC production by active layer samples throughout 350 days incubation exceeded gaseous C loss up to 23-fold under anoxic conditions, whereas production by permafrost peat was small. Taken together, our study suggests that permafrost peat in thawing Norwegian peat plateaus degrades at rates similar to those of active layer peat, while highest $CH_4$ production can be expected after inundation of thawed permafrost material in thermokarst ponds.

## 1 Introduction

In the northern hemisphere, around 15 % of the terrestrial surface is underlain by permafrost, of which 8 - 12% are permafrost-affected peatlands, covering ~1.7 million $km^2$ (Obu et al., 2019; Hugelius et al., 2020). About one third of the carbon (C) stored in permafrost affected soils is contained in peatlands (Lindgren et al., 2018), amounting to ~185 Pg C (Hugelius et al., 2020). In northern Scandinavia, peatlands have acted as a long-term C sink in the Holocene (Panneer Selvam et al., 2017). Rapid global warming, particularly in Arctic regions, is expected to cause permafrost thawing and destabilise carbon, leading to a net release of C into the atmosphere in

form of carbon dioxide ($CO_2$) and methane ($CH_4$) (Ramage et al., 2024; Wang et al., 2022) or to water courses in the form of dissolved organic carbon (DOC). Both $CO_2$ and $CH_4$ are greenhouse gases that contribute to global warming, and permafrost thawing may thus amplify global warming (Knoblauch et al., 2018).

Peatlands in northern Norway are located in the sporadic permafrost zone, forming peat plateaus and palsas, i.e. peat uplands and mounds with a frozen core lifted above the water table through formation of segregation ice (Alewell et al., 2011). Peat plateaus in northern Norway have decreased in lateral extent by 33-71% from 1950 to 2010, with the largest change recorded in the last decade (Borge et al., 2017). In permafrost peatlands, thawing often occurs abruptly by thermal erosion due to excess ice melt (Martin et al., 2021), exposing thawing permafrost

peat to anoxia when inundated in thermokarst ponds. Environmental factors controlling the degradation of organic matter from thawing peat plateaus are still poorly understood. Thermokarst ponds are the natural succession to peat plateaus after thawing and are thus crucial for understanding future permafrost degradation and climate feedback. Thermokarst ponds accumulate new peat over time, eventually forming non-permafrost peatlands (Clymo and Hayward, 1982). It is unclear whether the climate change driven transformation from peat plateaus

to thermokarst and non-permafrost peatlands results in an overall increase of C storage or whether the system is turned into a net C source (Turetsky et al., 2007; Treat et al., 2015; 2021). Understanding the mechanisms and extent of C degradation in thawing peat permafrost is therefore crucial for predicting future arctic C balances.

In this study, we characterised post-thaw degradation kinetics of peat from three Norwegian permafrost peatlands. The peat plateaus were selected to represent an area with sporadic permafrost differing in peat and permafrost

age, as well as in climatic conditions. We incubated samples from vertical peat profiles ranging from the surface through the active layer to the mineral layer below the permafrost peat and evaluated relative C degradation potentials of permafrost peat differing in age and decomposition history. We used both short- and long-term incubations at moderate temperature (10°C); short-term incubations to explore the immediate metabolic response with and without oxygen ($O_2$), as little is known about the resuscitation kinetics of microbial decomposers after

controlled thawing of permafrost peat; long-term incubation to evaluate differences in degradability of active layer and permafrost peat which have been shown to differ greatly with $O_2$ availability and temperature (Kirkwood et al., 2021; Treat et al., 2014; Waldrop et al., 2021; Panneer Selvam et al., 2017). Permafrost peat differs distinctly in $O_2$ availability between the active layer and the permafrost, with the former being drier and more exposed to $O_2$. However, $O_2$ availability in the active layer varies greatly during the year since it is frozen in winter and thaws

over the summer (Åkerman and Johansson, 2008). The collapse of peat plateaus and sequential formation of thermokarst ponds also changes the $O_2$ availability and thus the degradation potentials (Hodgkins et al., 2014). To compare freshly thawed permafrost peat with in situ thawed thermokarst peat, we incubated additional samples from corresponding depth profiles in neighbouring thermokarst ponds. As depth-resolved measurements are extremely rare for permafrost peat, the goal was to explore whether depth patters are site-specific or can be

generalized across permafrost peatlands in Northern Norway. The main objective of the study was to quantify the C degradation potentials of permafrost peat and its partitioning into $CO_2$, $CH_4$ and DOC. To study the influence of $O_2$ availability on C partitioning, peat samples were incubated both initially oxically and throughout anoxically for up to 350 days. Additionally, to explore the role of microbial growth in C degradation and mobilisation, a parallel set of samples were incubated as stirred soil slurries for 96 days, thus eliminating diffusional constraints

on substrate availability. To compare $CO_2$ and $CH_4$ production potentials along a thaw gradient, additional

incubations were carried out with recently thawed peat material obtained from shallow thermokarst depressions adjacent to two of the peat plateaus.

## 2 Materials and Methods

### 2.1 Site description

Three peat plateaus in the sporadic permafrost zone of Finnmark, northern Norway, were selected along a climatic gradient spanning from the coastal site Lakselv with maritime climate to the more continental sites Iškoras and Áidejávri (Table 1). Iškoras and Áidejávri are situated on Finnmarksvidda, a 22 000 km$^2$ plateau with elevations of 300 to 500 m a.s.l. At Iškoras, the peat development started around 9200 cal. yr. BP (Kjellman et al., 2018), while the peatland age at Áidejávri is unknown. However, the general timing of peat development is likely similar

to that at the Iškoras site. The peatland at the coastal site Lakselv developed around 6150 cal. yr. BP, after the site had emerged from the adjacent fjord due to postglacial land heave (Kjellman et al., 2018). Permafrost formation for Iškoras started during the Little Ice Age, with radiocarbon dates from a single core suggesting permafrost formation around 800 cal. yr. BP while formation for Lakselv occurred later around 150 cal. yr. BP (Kjellman et al., 2018). The peatlands in Lakselv are situated under the marine limit and the total column integrated C content

is smaller than at Iškoras (Kjellman et al., 2018). The dominant vegetation at the peat plateaus has been characterised as dwarf shrubs, mosses, lichen and cloudberry herbs at Iškoras and Lakselv (Kjellman et al., 2018; Martin et al., 2019). Thermokarst and surrounding wet fen areas are dominated by sedges, cotton grass and Sphagnum sp. (Kjellman et al., 2018; Martin et al., 2019). Áidejávri has similar vegetation, however, this site has not been investigated and described in detail. The soils at the three peat plateaus are characterised as histosols

(IUSS Working Group WRB 2014). Typical bulk densities in peat plateaus of the region range from 0.08 to 0.28 g cm$^{-3}$ in the active layer and permafrost peat, with low values below the active layer due to high excess ice contents and low values in the top of the active layer due to high porosity in the surface peat (Kjellman et al., 2018). This is also reflected in the gravimetric water contents in our thawed samples provided in table S11.

**Table 1: Location and characteristics of the three study sites. The mean annual air temperature (MAAT) and mean**
**annual precipitation (MAP) for period 1991-2020 are from the closest meteorological station (Šihččajávri for Áidejávri, Čoavddatmohkki for Iškoras, Banak for Lakselv) and were obtained from the Klimaservicesenter (2021). Differences in altitude were corrected using a standard lapse rate of −0.65°C/100 m. Peat and permafrost formation dates are from Kjellman et al. (2018).**

|  | Áidejávri | Iškoras | Lakselv |
|---|---|---|---|
| Coordinates | 68°44'59" N 23°19'06" E | 69°20'27" N 25°17'44" E | 70°7'14" N 24°59'47" E |
| Elevation (m a.s.l.) | 398 | 381 | 50 |
| MAAT (°C) | -2.0 | -1.9 | +1.4 |
| MAP (mm) | 478 | 433 | 392 |
| Peat formation (cal. yr. BP) | Not available | 9200 | 6150 |
| Permafrost formation (cal. yr. BP) | Not available | 800 | 150 |

## 2.2 Field sampling

Intact cores from the permafrost-underlain peat plateaus were collected in September 2020 at the time of maximum active layer depth. The active layer (AL) was sampled using a cutting tool and a small shovel. Each core was divided into three depth layers denoted as active layer (AL), transition zone (TZ, i.e. the top of the frozen layer, which is likely to thaw occasionally) and permafrost (PF) as shown in Fig. 1. For two of the three cores a fourth, mineral layer, was sampled below the peat. All active layer samples were kept cool until placed in a refrigerator at 3.8°C (SD=0.47°C) in the laboratory. At the time of sampling, the active layer depth was 0.6 m at Iškoras and Lakselv and 0.5 m at Áidejávri. The frozen TZ and PF cores were sampled using a steel pipe (outside diameter 38 mm, inside diameter 30 mm) that was hammered vertically in ~5 cm increments into the frozen peat. A paper towel was pushed through the pipe between each sample and tools used for handling samples were wiped with disinfectant to minimise cross-contamination between samples. Incremental coring was continued until reaching the mineral soil below the peat. The total thickness of the organic core (active and frozen peat layers) was 1.67 m at Iškoras, 1.04 m at Áidejávri and 0.85 m at Lakselv. Sub-samples of the core (~5 cm in length) were transferred to 50 ml centrifuge tubes, which were immediately sealed by a screw cap and placed into a freezing box kept below 0°C until being transferred to a -20°C freezer on the same day. The samples were shipped frozen to the laboratory, where they were kept at -17.8°C (SD = 0.39°C). To facilitate comparison between permafrost cores from different sites, the samples were assigned to seven operational layers according to their vertical structure, consisting of (from top to bottom) three samples from the active layer (AL1, AL2, AL3), one from the transition zone (TZ) and three from the permafrost layer (PF1, PF2, PF3). The active layer samples were assigned upon a visual inspection. AL1 and AL3 were sampled from the top and bottom of the active layer, respectively, and AL2 from the middle. AL1 did not contain surface vegetation but was less decomposed than AL2 and AL3. The TZ sample was taken from the top of the frozen core and the PF1 sample just below this. PF3 was taken from the bottom of the frozen core which consisted of mineral soil at Áidejávri and Lakselv. PF2 samples were taken in between PF1 and PF3. The absolute depth of the operational layers differed across the three permafrost cores (Table S1).

To study the C degradation potential of recently thawed and inundated peat material, thawed peat material from thermokarst depressions adjacent to the peat plateaus was sampled in September 2021 and incubated for 96 days (see below). Using orthophotos from annual drone overflights, the time of thawing was estimated to 2017/2018 for Iškoras (Fig. 2, similar to Martin et al. (2021)) and to before 2003 (from earliest available orthophoto, Norgeibilder (2023), Fig. 3) for Áidejávri. Visual inspection of the thermokarst cores allowed to distinguish the more decomposed former active layer peat (TK-AL) from the former permafrost peat (TK-PF1 and TK-PF2/3) below (Fig. 1), so that samples corresponding to the layers distinguished in the permafrost cores (Table S2) could be taken. In Iškoras, the former peat plateau surface was submerged under 20 cm of standing water, while a fresh peat layer had formed on top of the submerged peat plateau surface at the Áidejávri site.

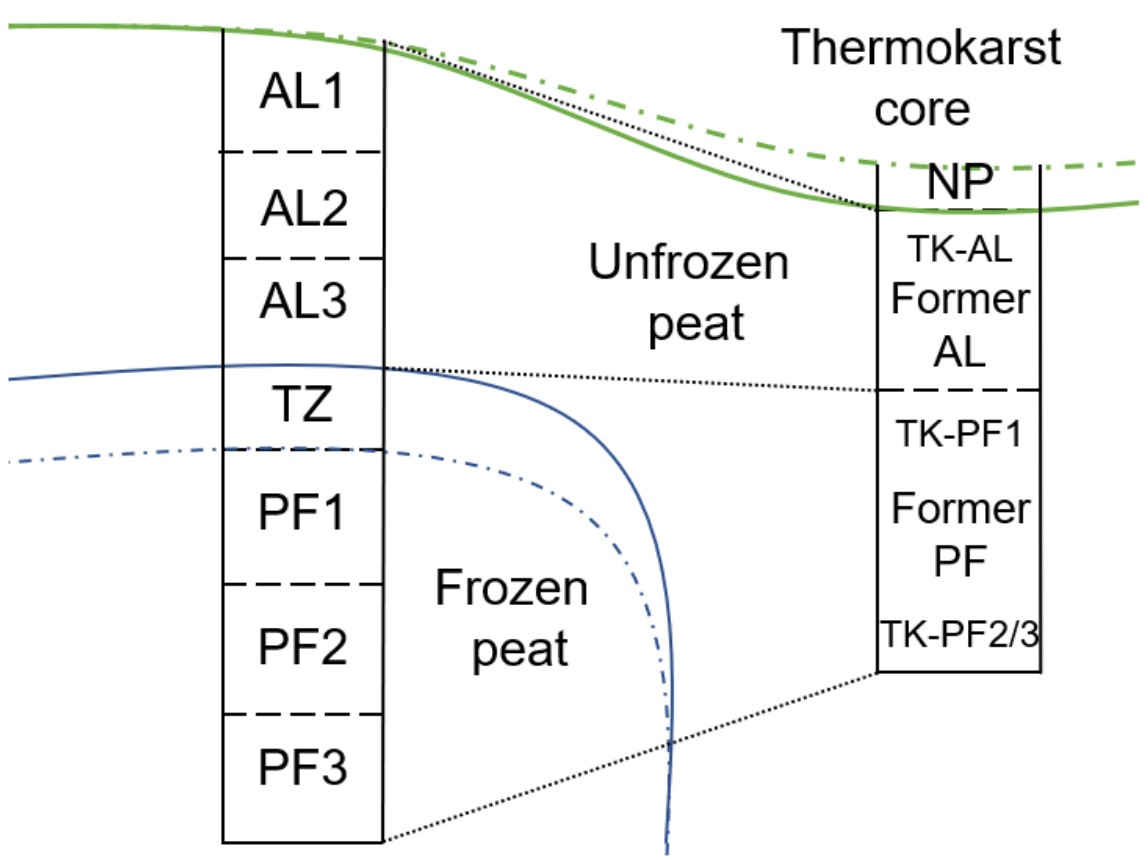

140

**Figure 1: Schematic drawing of permafrost and thermokarst core. The green line indicates the surface, while the green dashed line indicates growth of new peat. The blue line shows the approximate position of the permafrost table. The blue dashed line is the maximum thaw depth which can be occasionally thawed. AL = active layer, TZ= transition zone, PF = permafrost, TK = thermokarst, NP = new peat (i.e. accumulated following thaw, only at Áidejávri).**

145

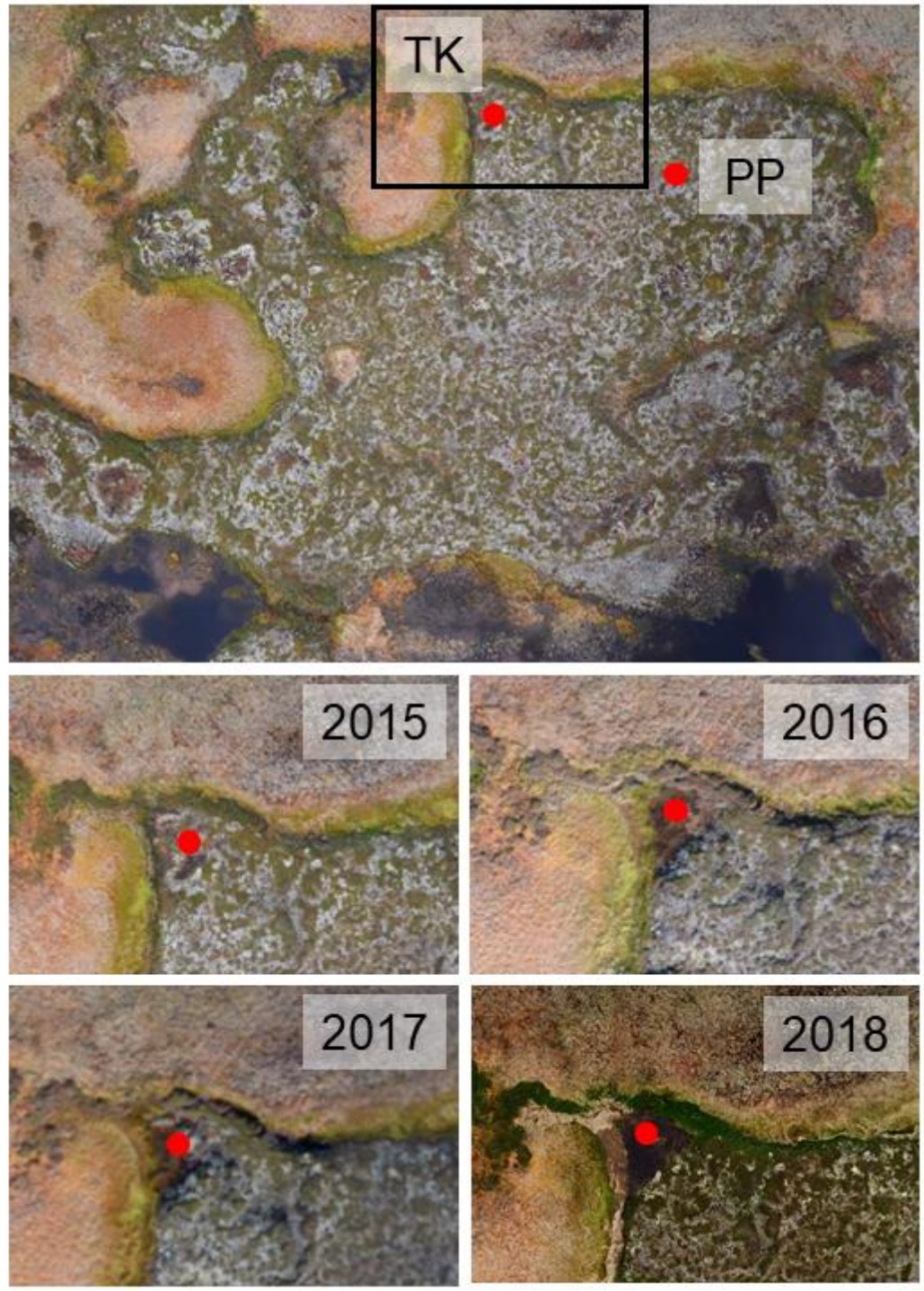

**Figure 2: Aerial images of the Iškoras sampling site obtained by drone (processing as in Martin et al., 2021). Top: 2015 orthophoto from a drone survey (scene width 100m), with red dots showing the locations of peat plateau (PP) and thermokarst (TK) cores sampled in 2020 and 2021, respectively. Bottom four: magnification of black rectangle for repeat orthophotos from September 2015-2018, showing the initially intact peat plateau edge collapsing in a shallow thermokarst pond from where thermokarst samples were taken in 2020.**

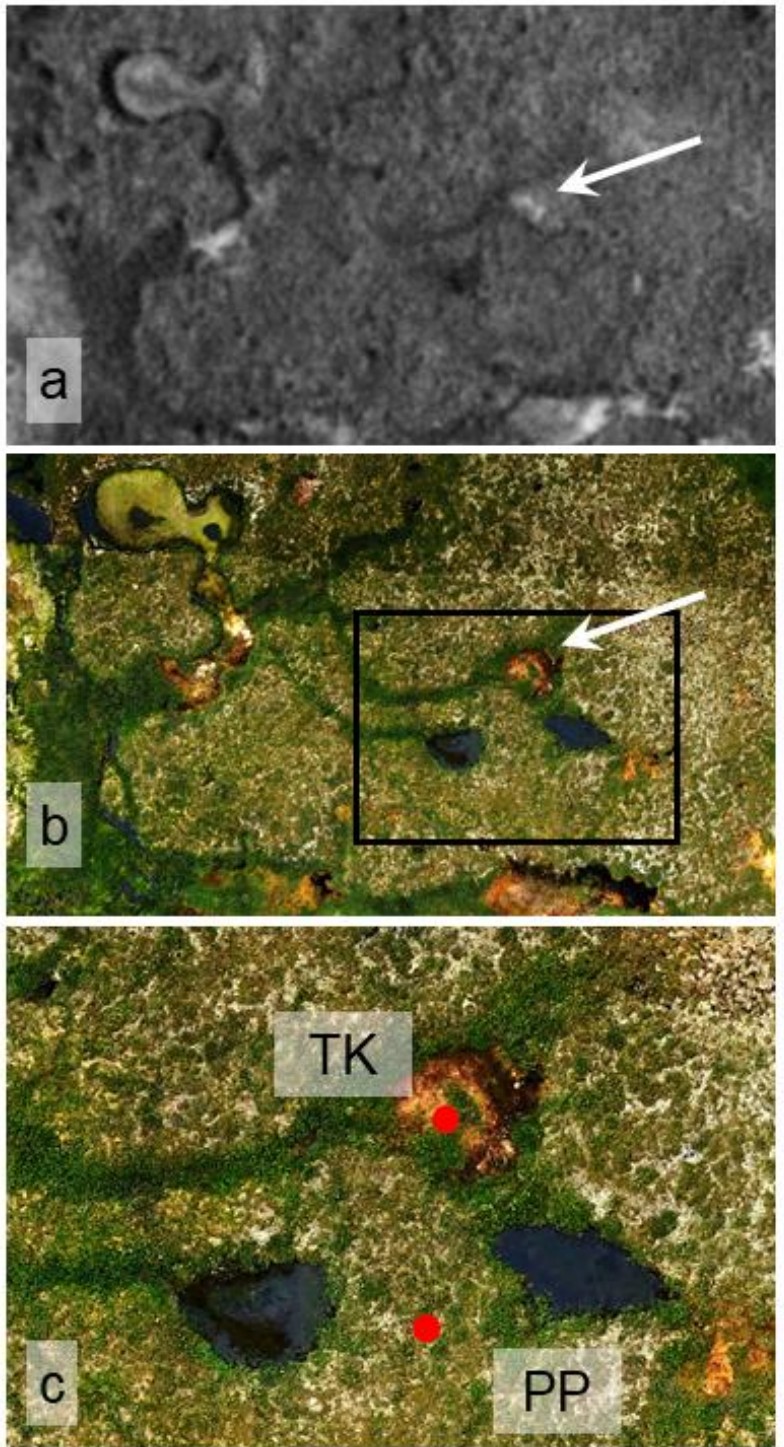

**Figure 3: Aerial images of the Áidejávri sampling site; a) 2003 aerial image by the Norwegian Mapping Authority (Norgeibilder, 2023); b) 2018 orthophoto from a drone survey (processing as in Martin et al., 2021); scene width a, b: 130m; c) magnification of black rectangle in b, with red dots showing the locations of peat plateau (PP) and thermokarst (TK) cores; white arrow pointing to drained thermokarst pond visible in both 2003 and 2018 imagery.**

## 2.3 Stable isotope signatures and elemental analysis

Elemental composition was analysed on freeze-dried, homogenised samples using ICP-MS (Agilent Technologies 8800 ICP-MS Triple Quad) and ICP-OES (PerkinElmer FIMS). First, 0.20-0.25 g of the homogenised, freeze-dried material was weighed in acid-washed Teflon tubes. Each tube was filled with 2 ml ultrapure $dH_2O$ and 5 ml ultrapure concentrated $HNO_3$. Samples were incubated overnight to ensure moisturisation, before decomposition

in an ultraclave (ultraCLAVE, Milestone). The decomposed sample was then added to a fresh 50 ml centrifugation vial, amended with 20 ml ultrapure $dH_2O$ and 1 ml concentrated HCl before filling up the tube to 50 ml with ultrapure $dH_2O$. The tubes were shaken 10 times to ensure mixing. Traceability and accuracy of the elemental analysis were ensured by including standards of known elemental composition (NCS ZC73013, NCS DC73349, Peach Leaves-1547, Pine Needles-1575 and River Sediment-LGC6187).

Contents of C and N and their natural $^{13}C$ and $^{15}N$ abundances were analysed using a flash combustion elemental analyser (Thermo EA 1112 HT O/H-N/C) coupled through a ConFlow3 interface to an isotope ratio mass spectrometer (Thermo-Finnigan Delta Plus XP). Isotope values were calibrated against certified reference materials (IAEA-N1, IAEA-CO8) and are expressed in delta ($\delta$) notation relative to VPDB and atmospheric $N_2$, respectively. To compare C content with organic matter content, loss on ignition was measured at 550ºC ±25 with samples dried at 60ºC overnight.

## 2.4 Incubation set-up

Incubations were set up in batches, studying one complete core of each site at a time without technical replication. Each permafrost core was divided into seven layers (Fig. 1, Table S1) and the thermokarst core was divided into three or four layers (Table S2) from which four samples each were prepared. Each sample was divided into four portions which were placed in 120 ml serum bottles and capped with crimp-sealed Butyl septa. Frozen core samples (from TZ and PF layers) were divided lengthwise while being frozen to minimise gas release. To remove gases released during sample preparation and to minimise exposure to $O_2$ during thawing, the bottles were placed on ice and washed with Helium 6.0 (He) using an automated manifold alternately evacuating and filling the headspace 7 times. Helium overpressure was removed before placing the bottles into a temperature-controlled cabinet at 3.8°C for overnight thawing. Headspace gas concentrations were determined after ≥20 h thawing.

After controlled thawing in He and measuring gas release, the four replicate bottles for each sample were assigned to different treatments: two bottles were kept as 'loose peat' at natural moisture content (Table S11), while the peat in the other two bottles was dispersed in 52 ml ultra-distilled water (3.8 °C) by magnetic stirring, creating 'peat slurries'.

The slurries were stirred for one hour to fully disperse the peat, whereafter the peat material was allowed to settle, and 2 ml supernatant was sampled with a syringe to measure pH and DOC concentration. pH was measured using a HACH H170 pH meter, before centrifuging the aliquot at 10,000 G for 10 minutes and filtering it through a 0.45 µm filter (Sterile Syringe Filter with polyethersulfone membrane, VWR International) for analysing water-extractable DOC by a Total Organic Carbon Analyser (TOC-V, Shimatzu, Japan). Hereafter, one bottle of each set (loose and slurry) was washed with He as described above to create anaerobic conditions, while the other two were washed with a He/$O_2$ (80/20) mixture to create initially aerobic conditions. A major goal of this study was to explore the depth-dependency of peat mineralisation rates. Given the constraints of the laboratory setup (e.g. number of bottles which could be incubated simultaneously) and the limited sample volume available per depth, we did not combine the samples of bulk layers (e.g. the entire active layer) and perform technical replicates (as in e.g. Kirkwood et al., 2021; Treat et al., 2014). Instead, we retained the high depth resolution and performed only a single incubation per depth and treatment. While this can lead to higher uncertainty in individual incubation results, we generally base our findings on a number of incubated samples (e.g. full vertical depth profiles), thus strongly moderating the influence of single biased measurements.

## 2.5 Incubation experiments

All bottles, both the loose and the stirred peat suspensions, were incubated in a temperature-controlled water bath at 10°C using an incubator with automated gas analysis (Molstad et al., 2007; Molstad et al., 2016). The incubator consists of a temperature-controlled water bath with submersible stirring boards holding up to 44 serum bottles (120 ml) and is placed under the robotic arm of an autosampler (GC-Pal, CTC) which repeatedly pierces the septum of the bottles with a hypodermic needle and pumps ~1 ml via a peristaltic pump (Gilson 222XL) to a multi-column, multi-detector gas chromatograph (Agilent 7890A) equipped with an automatic sample admission system. Upon injection, the peristaltic pump is reversed, and sample gas not injected onto the columns is pumped back to the bottle together with He, maintaining the pressure in the bottles at ~1 atm. The GC has two columns, a poraplot Q column to separate $CH_4$, $CO_2$ and $N_2O$ from bulk air and a Molesieve column to separate $O_2+Ar$ from $N_2$ and three detectors (TCD, FID, ECD) for simultaneously determining $O_2$, $CO_2$, $N_2$, $CH_4$, and $N_2O$ concentrations. Dry bottles with standard mixtures of known concentrations (AGA, Norway) were included into the measurement sequence for calibration and for evaluating the dilution resulting from back-pumping He after each sampling. He-filled bottles were included to evaluate leakage of $O_2$ into the measurement system. After converting peak areas to ppmv, moles of $CO_2$ and $CH_4$ accumulated or $O_2$ consumed were calculated taking account of dissolution in peat water (Wilhelm et al., 1977; Appelo and Postma, 1993), dilution by He back-pumping and leakage of $O_2$ during sample admission. For more details, see Molstad et al. (2007).

Headspace gas concentrations were monitored every 4.5 h for 413 to 450 h (~17 to 19 days) for permafrost cores and 180 h (~7 days) for thermokarst cores to investigate post-thaw gas kinetics at high resolution. Thereafter, the bottles were transferred to a temperature-controlled cabinet adjusted to 9.7°C (SD=0.04), where the incubations were continued without stirring. The slurries were shaken at least once a week, after which headspace samples were retrieved manually for offline gas chromatography. After about one month, the measurement frequency was decreased to biweekly until 96 days of incubation upon which the incubation of slurries and thermokarst samples was discontinued. Loosely packed permafrost core samples continued incubation at 10°C until 350 days after thawing, sporadically measuring headspace concentrations (after ~10.5 months and ~1 year).

To evaluate microbial growth from initial high-resolution gas kinetics, we fitted selected periods of exponential product accumulation to a growth model (Eq. 1) following Stenström et al. (1998; 2001):

$$p = p_0 + \frac{r}{\mu} * (e^{\mu t} - 1) + Kt \qquad (1)$$

Where $p$ is the product concentration, $p_0$ the product concentration at time 0, $r$ the total respiration rate, $\mu$ the specific growth rate, $t$ the time and $K$ a constant respiration rate by nongrowing microorganisms. Before fitting the curves, we truncated $CO_2$ kinetics from the first 17 to 19 days of incubation, excluding the transitional phase of decreasing accumulation rates towards the stationary phase. Sigmaplot 14.0 was used to fit Eq. 1 to the selected data and significant ($p< 0.05$) specific growth rates ($\mu$) are reported. A student's t-test with two tailed distribution and two sample unequal variance was used to evaluate differences in geochemical peat characteristics among the sites using Microsoft Excel for Microsoft 365.

## 3 Results

### 3.1 Peat characteristics

The peat plateau at Iškoras featured the highest C content among the three studied permafrost cores, with little variation throughout the peat profile (491-552 mg C g dw$^{-1}$, Fig. 4). Permafrost cores from Áidejávri and Lakselv had similar C contents in the active layer (439-551 mg C g dw$^{-1}$) but held less C in the permafrost. Organic matter (OM) content mirrored C content, except for the deep permafrost (PF3) at Lakselv which indicated high concentration of inorganic C in this layer (Fig. 4). Other layers showed no indication of inorganic C and C contents

are henceforth referred to as organic C (OC). The permafrost core from Iškoras had the highest C/N ratios irrespective of depths, with the highest ratio in the top active layer (AL1) gradually decreasing with depth. Áidejávri and Lakselv also featured highest C/N ratios in AL1, below which they decreased more strongly throughout the active layer, while being stable throughout the permafrost zone. DOC extracted 20 h after thawing was highest in samples from the transition zone and from the top of the permafrost zone, with values at Iškoras

clearly exceeding those at Áidejávri and Lakselv (p<0.019). pH increased with depth in all three cores and was significantly lower (p<0.009) at Iškoras compared to the other two cores. Lakselv was most depleted in $^{13}$C, whereas Iškoras and Áidejávri showed more variable δ$^{13}$C values with depth. As with δ$^{13}$C, the δ$^{15}$N values were highest in the top layer (AL1) and decreased throughout the active layer, most at Iškoras, and least at Lakselv. Áidejávri and Lakselv featured the highest iron (Fe) contents, while Iškoras had almost no Fe throughout the

profile. Phosphorus (P) content was quite even throughout the profiles at all sites, while Lakselv had a higher P content than the other two sites. Sulfur (S) content increased in the TZ and PF layers at Iškoras and Áidejávri, while it was more evenly distributed with depth at Lakselv (Fig. 4).

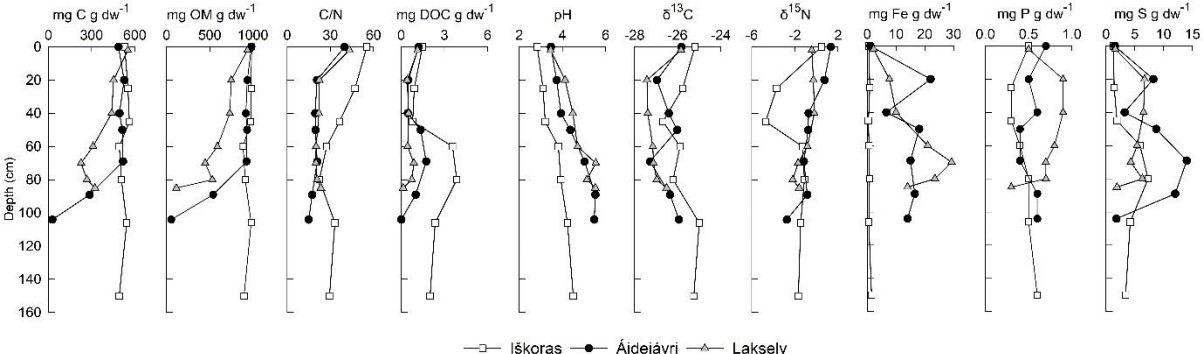

**Figure 4: Depth profiles of geochemical variables in permafrost cores at Iškoras, Áidejávri and Lakselv. Shown are (from left to right) carbon content (mg C g dw$^{-1}$), organic matter content (mg OM g dw$^{-1}$), carbon to nitrogen ratio (C/N), dissolved organic carbon (mg DOC g dw$^{-1}$), pH, isotope signatures of carbon (δ$^{13}$C) and nitrogen (δ$^{15}$N), iron content (mg Fe g dw$^{-1}$), phosphorous content (mg P g dw$^{-1}$) and sulfur content (mg S g dw$^{-1}$). The deepest layer at Áidejávri and Lakselv were affected by mineral soils. The thaw depths at the coring location were 60 cm (Iškoras and**

**Lakselv) and 50 cm (Áidejávri).**

### 3.2 Gas kinetics

The initial gas formation throughout the first 20 days revealed clear kinetic differences between sites and depths in permafrost cores as showcased in Fig. 5 and Table 2. CO$_2$ production kinetics were most dynamic for Iškoras

(TZ) showing exponential product accumulation during initial incubation (0-19 days) of permafrost samples both under oxic and anoxic conditions (e.g. insert in Fig. 5a; Table 2). A similar pattern was observed for samples from Áidejávri and Lakselv (Fig. 5a, Table 2). By contrast, all AL samples incubated loosely showed linear $CO_2$ accumulation (Table 2).

When incubating samples as stirred slurries, two effects were observed: some of the AL samples turned to
exponential $CO_2$ accumulation and there was a tendency of greater apparent microbial growth as indicated by specific growth rates, $\mu$, in Table 2, except for anoxically incubated samples from Áidejávri and Lakselv. Estimated specific growth rates ranged from 0.07 to 0.007 $h^{-1}$, which corresponds to generation times of 14 to 145 hours. $CO_2$ production at all sites and depths showed a tendency to level off over time (Fig. 5a).

Except for Áidejávri, $CH_4$ accumulation during the first 20 days was small and curve-linear while rates in samples
from Lakselv and Áidejávri did not increase before after ~100 days of incubation. During the first 96 days of the incubation, all Iškoras PF and TZ samples displayed $CH_4$ kinetics as shown in Figure 5b, with apparent initial $CH_4$ production levelling off over time no matter whether the samples were incubated stirred or unstirred, oxically or anoxically (Fig. S1), suggesting that initial $CH_4$ release was not due to methanogenesis. This was further supported by the higher initial $CH_4$ release from Iškoras samples compared to Áidejávri during the first 20 hours
of thawing (Table S6). In contrast, TZ, PF1 and PF2 samples from Áidejávri accumulated $CH_4$ exponentially throughout the first 19 days, indicating methanogenesis (Fig. 5b, Fig. S1). After levelling off, $CH_4$ production increased in PF and TZ samples from Áidejávri after about 80 days and continued throughout the remainder of the incubation. Samples from Lakselv featured less $CH_4$ production in the beginning of the incubation which, however, increased greatly towards the end of the incubation, resulting in greater final $CH_4$ accumulation than in
samples from Áidejávri or Iškoras (Fig. 5b). $N_2O$ kinetics (production and uptake) were observed but accumulated amounts were minute and are not reported here.

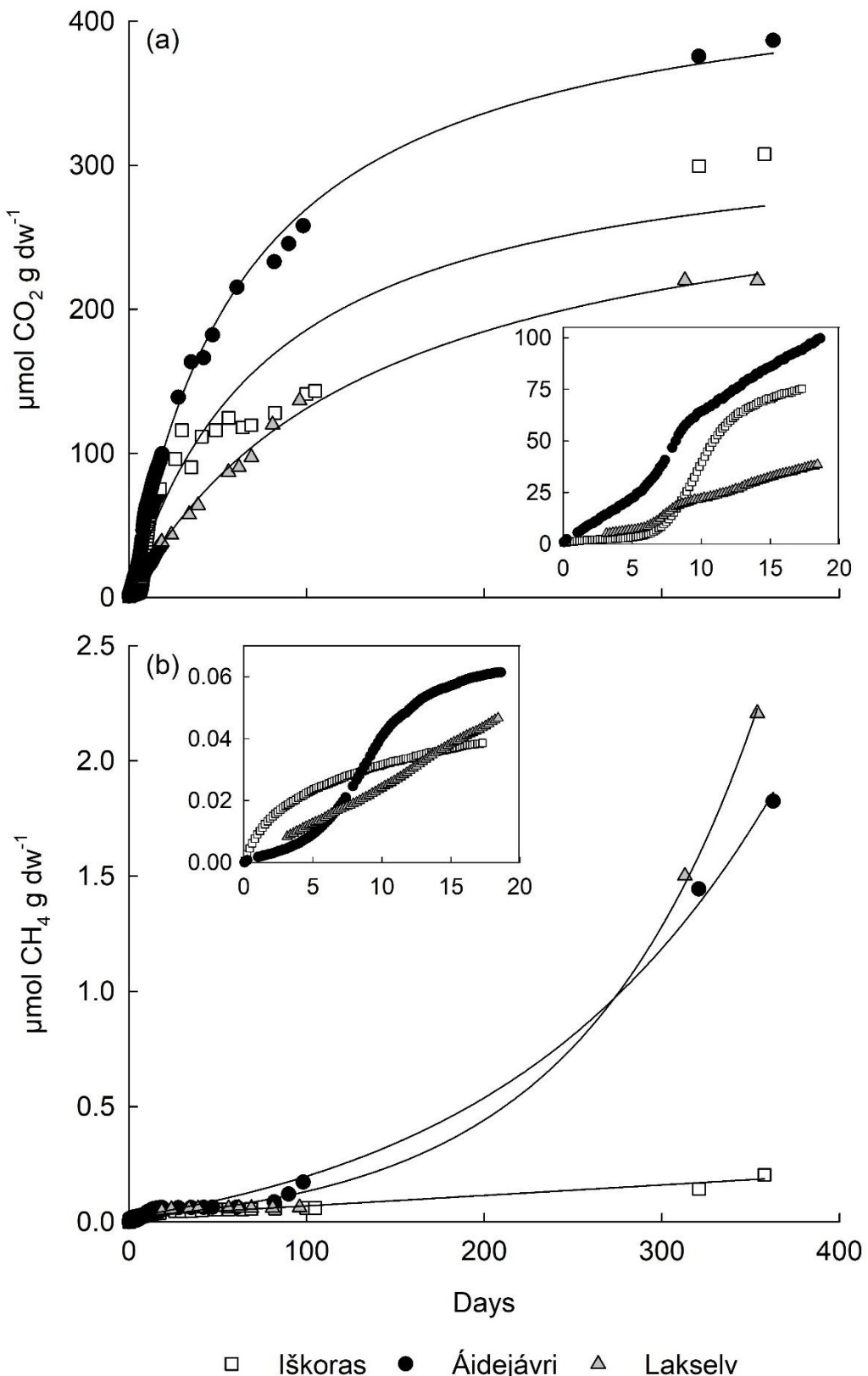

**Figure 5: Kinetics of $CO_2$ and $CH_4$ release from permafrost core samples (loosely packed) from layers with highest $CH_4$ production at Iškoras (TZ), Áidejávri (PF1) and Lakselv (PF2). Initial data for Lakselv are missing due to instrument failure. (a) $CO_2$ accumulation of initially oxic peat throughout 350 days with fitted hyperbola ($f = a*x/(b+x)$). (b) $CH_4$ accumulation of anoxically peat throughout 350 days fitted to Eq. 1.**


**Table 2: Specific growth rates (h$^{-1}$) derived from the exponential part of high-resolution CO$_2$ kinetics during initial incubation (0 to 19 days). All rates are statistically significant at p<0.05. Averages are calculated for TZ and PF samples ± SD.**

| Treatment/ Layer | Iškoras | | Áidejávri | | Lakselv | |
|---|---|---|---|---|---|---|
| | Oxic | Anoxic | Oxic | Anoxic | Oxic | Anoxic |
| Loose | | | | | | |
| AL1 | - | - | - | - | - | - |
| AL2 | - | - | - | - | - | - |
| AL3 | - | - | - | - | - | - |
| TZ | 0.020 | 0.026 | 0.028 | - | - | 0.007 |
| PF1 | 0.020 | 0.030 | 0.027 | 0.070 | - | 0.058 |
| PF2 | 0.026 | 0.031 | 0.034 | 0.070 | 0.028 | 0.040 |
| PF3 | 0.039 | 0.022 | 0.039 | 0.027 | 0.032 | - |
| Average TZ and PF layers | 0.026 ± 0.008 | 0.027 ± 0.004 | 0.032 ± 0.005 | 0.056 ± 0.02 | 0.03 ± 0.002 | 0.035 ± 0.021 |
| Slurry | | | | | | |
| AL1 | - | 0.047 | - | - | - | - |
| AL2 | 0.039 | 0.023 | - | 0.063 | - | 0.047 |
| AL3 | - | 0.051 | - | 0.066 | - | 0.012 |
| TZ | 0.047 | 0.056 | - | - | - | 0.008 |
| PF1 | 0.025 | 0.063 | 0.054 | 0.023 | 0.065 | 0.012 |
| PF2 | 0.027 | 0.024 | - | - | 0.040 | - |
| PF3 | 0.030 | 0.022 | - | - | - | - |
| Average TZ and PF layers | 0.032 ± 0.009 | 0.041 ± 0.019 | 0.054 | 0.023 | 0.053 ± 0.012 | 0.01 ± 0.002 |

## 3.3 Cumulative CO$_2$ production

Expressed as cumulative CO$_2$ production over 96 days, the AL1 layers of all three permafrost cores showed largest CO$_2$ production under oxic conditions (Fig. 6). Cumulative CO$_2$ production decreased strongly with depth throughout the active layer, before increasing again in the TZ layer and reaching a secondary maximum in the upper permafrost layer. Maximum CO$_2$ production of PF peat after 96 days accounted for 42, 102 and 60% of the CO$_2$ production observed in the top of the active layer (AL1) at Iškoras (PF1), Áidejávri (PF1) and Lakselv (PF2), respectively, indicating that oxic post-thaw respiration of permafrost peat can reach values comparable to those of the surface layer with fresh litter input. Across the three sites, PF peat at Lakselv accumulated least CO$_2$ (Fig. 6). Incubating initially aerobic samples beyond 96 days revealed a gradual decrease in CO$_2$ production in all layers of all sites (Fig. 5a), which coincided with O$_2$ depletion and in some cases even O$_2$ exhaustion (Fig. S4, S6 and S8). Notwithstanding, AL and PF top samples of the initially oxic treatment showed the largest relative increments in cumulative CO$_2$ production from 96 to 350 days (Fig. 6). After 350 days, CO$_2$ accumulation of PF samples from Iškoras (PF1) reached 67 % of the CO$_2$ accumulation in the AL1 sample, Áidejávri (PF1) 125 % and Lakselv

(PF2) 72 %. $CO_2$ production was greatly reduced in the absence of $O_2$, irrespective of depth and incubation time (Fig. 6).

In general, there was little effect of stirring on the decomposition during the first 96 days (Fig. S3), indicating that peat degradation was little controlled by matrix effects or diffusional constraints. However, due to the addition of water and the associated decease in headspace volume, there was overall less $O_2$ available in slurried samples than in loosely packed peat, which likely shortened the period of oxic respiration (Fig. S4 to S9).

$CO_2$ production in thermokarst cores (Iškoras and Áidejávri) was in the same order of magnitude (40 to 241 µmol $CO_2$ g dw$^{-1}$ 96 days$^{-1}$) as by the permafrost cores but was exceeded by $CO_2$ production by new 'peat' at Áidejávri (616 µmol $CO_2$ g dw$^{-1}$ 96 days$^{-1}$) (Table S3).

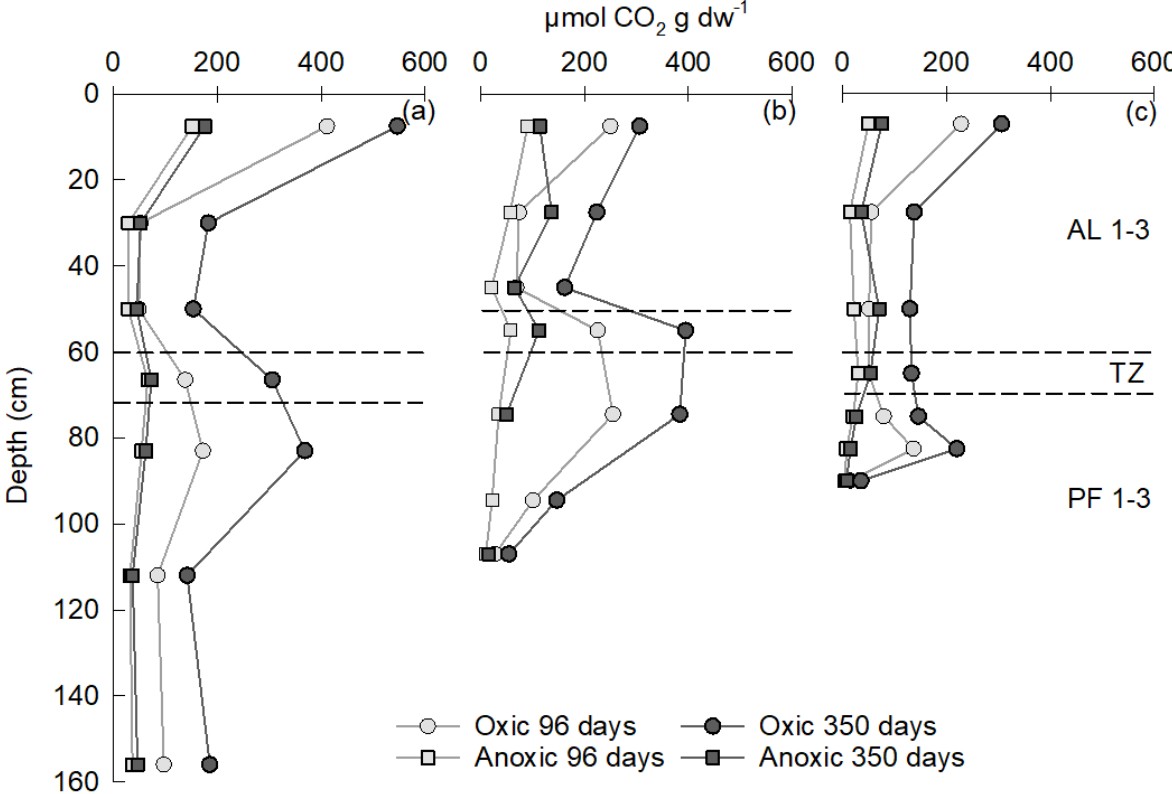

**Figure 6: Depth profiles of cumulative $CO_2$ production throughout 96 and 350 days of incubation as loosely packed permafrost core samples. (a) Iškoras, (b) Áidejávri and (c) Lakselv. The depth indicates the average depth of the incubated sample. Stippled lines indicate thaw depth at sampling and the location of the transition zone (TZ). Anoxic PF2 for Áidejávri could not be measured after 96 days due to leakage.**

## 3.4 Methane release and production

Significant $CH_4$ production was only observed in TZ and PF samples despite prolonged anoxic incubation of samples from all depths (Fig. 7). Methane production after 96 days was up to 4 orders of magnitude smaller than $CO_2$ production on a C basis. Yet, while $CO_2$ production slowed down over time, $CH_4$ production increased (Fig. 5b) indicating that methanogenesis in thawing PF peat had a lag phase before producing $CH_4$ (Knoblauch et al., 2018). Over 350 days, the anoxic $CH_4$-C accumulation at Lakselv (PF2) reached 38.6 % of its $CO_2$-C

accumulation, while the corresponding values for Áidejávri (PF1) and Iškoras (TZ) were 9.2 % and 0.4 %, respectively.

As with $CO_2$, suspending and initially stirring the peat slurry had little effect on cumulative $CH_4$ production within the first 96 days (Fig. S2). However, during the initial high-resolution gas measurements $CH_4$ production in PF1 of Áidejávri, the only layer showing biogenic $CH_4$ production during initial anoxic incubation, was inhibited by stirring. High-resolution $CH_4$ kinetics of the non-stirred sample were markedly sigmoid (Fig. S1), suggesting that stirring may inhibit growth of methanogens.

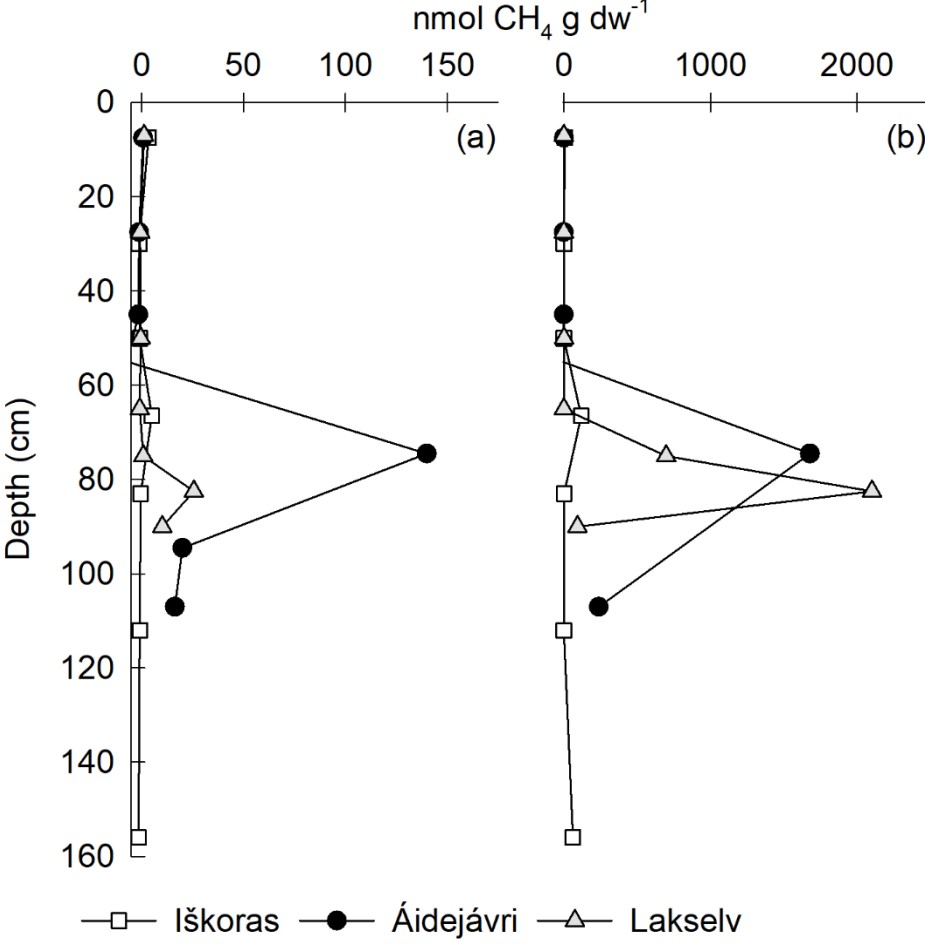


**Figure 7: Depth profiles of cumulative $CH_4$ production in permafrost cores from Iškoras, Áidejávri and Lakselv incubated anoxically as loosely packed samples. (a) 96 days, (b) 350 days. $CH_4$ was corrected for desorption by subtracting $CH_4$ accumulating in oxic bottles from anoxic $CH_4$ release to obtain biogenic $CH_4$ production (Fig. S2). Anoxic PF2 from Áidejávri could not be measured after 350 days due to leakage. The active layer depths at the coring**
**location were 60 cm (Iškoras and Lakselv) and 50 cm (Áidejávri).**

$CH_4$ production in thermokarst samples differed markedly from that of the corresponding layers in the permafrost cores (Table S4). At Iškoras, $CH_4$ production in thermokarst cores was between 2 and 4 orders of magnitude larger than in corresponding samples from the intact permafrost core after 96 days. Differences between thermokarst
and corresponding permafrost layers were somewhat smaller for Áidejávri, but still pronounced (Table S4). At both sites, $CH_4$ production was largest in the top layer of the thermokarst core (3 to 4 orders of magnitude larger than that in AL1 from the permafrost core), while $CH_4$ production of TK-PF1 and TK-PF2/3 at Áidejávri was in

the same order of magnitude (Table S4). In general, thermokarst samples responded more to $O_2$ and stirring/non-stirring than samples from the intact permafrost core, which makes it difficult to interpret differences between depths and sites (Fig. S11 and S12). Nevertheless, the potential to produce $CH_4$ increased dramatically in the former active layer peat when inundated (TK-AL) for both sites (Table S4), illustrating the strongly increased methanogenetic potential of peat plateau AL peat after thermokarst formation.

## 3.5 Total C mobilisation

Net release of DOC was measured as difference between initial and final extractable DOC. It greatly exceeded $CO_2$-C release in loosely packed active layer samples (Fig. 8). Initially oxically incubated samples (Fig. 8a) had markedly smaller DOC but larger $CO_2$-C production than anoxically incubated samples (Fig. 8b) and vice versa. pH increased more in anoxic incubations (Table S10) which might have made DOC more easily extractable. When combining net DOC release/uptake and $CO_2$-C production, total C mobilisation was largest in the active layer irrespective of initial $O_2$ status (Fig. 8). AL2 from Áidejávri showed exceptionally high C mobilisation compared with Iškoras and Lakselv. TZ and PF samples showed a tendency of net DOC consumption, which was most pronounced in samples from Iškoras and matches high apparent specific growth rates there (Table 2).

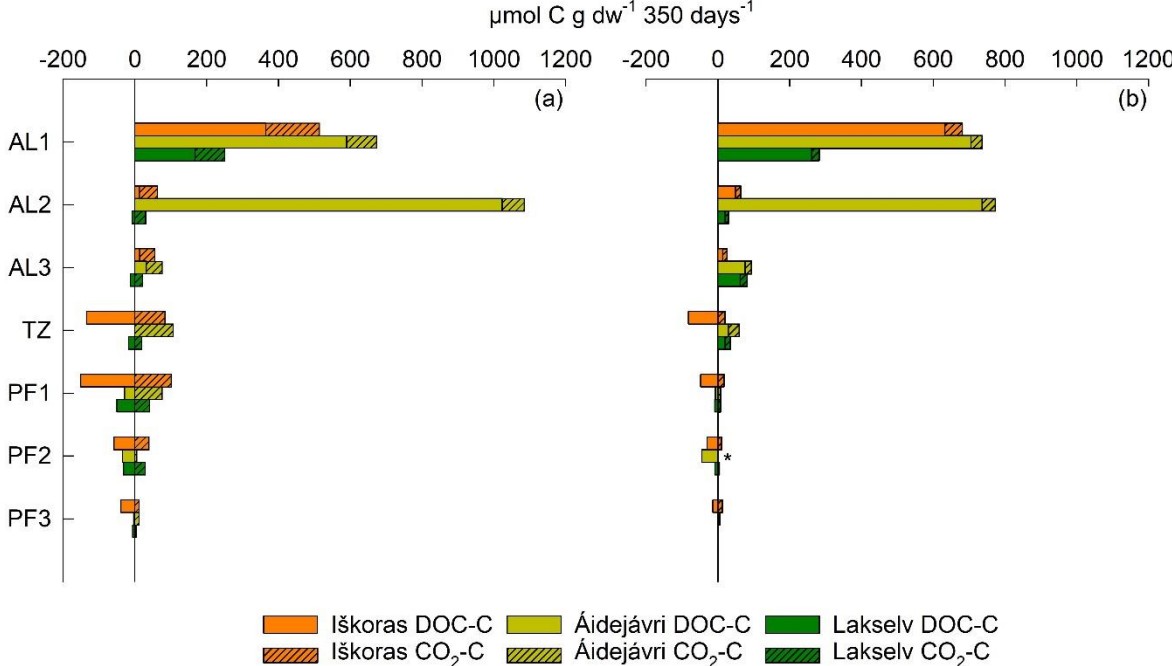

**Figure 8: Stacked CO₂-C production and net DOC release/uptake throughout ~350 days of incubation at 10 ºC. Data shown are from loosely packed permafrost core samples (a) incubated initially oxically and (b) anoxically. * no CO₂ data available due to leakage.**

## 4 Discussion

The cumulative $CO_2$ and $CH_4$ production of three permafrost cores showed a distinct depth pattern (Fig. 6 and 7). Largest $CO_2$ production was found in the top of the active layer, in the transition zone and in the top of the permafrost (Fig. 6). This suggests that thawed permafrost peat from Norwegian peat plateaus has a considerable potential for $CO_2$ production, comparable with that of active layers. By contrast, $CH_4$ production under anoxic

conditions was almost exclusively found in samples from thawed permafrost layers (Fig. 7). So far, only few studies have investigated decomposition in decaying peat plateaus by comparing post-thaw $CO_2$ production in thawed permafrost peat with that of overlaying active layers (Treat et al., 2014; Waldrop et al., 2021; Kirkwood et al., 2021; Harris et al., 2023).

Although caution is warranted when comparing decomposition rates across different incubation studies, Kirkwood et al. (2021) found, similarly to our study, highest $CO_2$ release in the top of the active layer for Canadian peat plateaus (see Supplement in Kirkwood et al. (2021)) and for some sites a second maximum in the permafrost peat. To compare the decomposition dynamics of the active layer and the permafrost peat in our study with those of Kirkwood et al., (2021), we calculated average rates for active layer and permafrost (Table S7). The comparison

reveals a similar trend in both studies with the active layer having higher degradation potentials under anoxic conditions than PF layers. The rates of anoxic $CO_2$ accumulation in active layer and permafrost peat reported by Kirkwood et al. (2021) were similar to the rates we found for Lakselv after adjustment for temperature and length of incubation (Table S7). Both Iškoras and Áidejávri featured higher $CO_2$ production rates in the active layer and the permafrost peat (Table S7) than the averages reported by Kirkwood et al. (2021). This could indicate a higher

C degradability at our sites in Northern Norway compared to the Canadian sites, but this should be corroborated by more detailed studies including technical replicates of the incubated samples.

By contrast, Treat et al. (2014) and Waldrop et al. (2021) did not observe differences in C degradation between different depths in Alaskan peat plateaus. However, these studies differed in peat stratigraphy from our study. Furthermore, the degradation potentials were higher than those found in the present study (Table S8 and S9).

Harris et al. (2023) found that C degradation in Canadian peat plateaus was highest in the top of the active layer and that deeper layers had low $CO_2$ production throughout. These variations could be due to differences in C quality among the different sites, or due to differences in sample treatment.

In general, incubation conditions differ across published studies and information about peat formation and quality is often lacking, making it difficult to compare C degradation rates. Our study demonstrates the variability of

potential degradation rates within the same geographical region, suggesting that more studies are needed to estimate potential climate feedback of permafrost peat thaw at regional scales. A standardised incubation protocol would improve the comparability among studies and help validating and improving numerical models of C dynamics in permafrost peatlands (e.g. Treat et al., 2021) which are critical tools to quantify present and future climate change impacts on these sensitive ecosystems.

**4.1 Constraints and variability in C decomposition**

Oxygen availability appeared to be the most important factor for C decomposition. $CO_2$ accumulation after 350 days in anoxic incubations reached only 7 to 61 % of that in initially oxic incubations (Table S12). This is in agreement with other studies, which found that $O_2$ availability is critical for $CO_2$ production after permafrost thaw (Estop-Aragones et al., 2018; Schädel et al., 2016). The incubation study by Waldrop et al. (2021) with arctic

Alaskan peat found that $CO_2$ accumulation after 6 months at 5°C under anoxic conditions accounted for 26% of that measured under oxic conditions. Also, initially oxic conditions did not appear to increase subsequent anoxic C degradation.

Incubating the samples as stirred slurries versus loosely placed peat affected initial gas kinetics. We observed exponential $CO_2$ accumulation within the first 20 days of incubation in some of the bottles, indicating microbial

growth (Table 2, Fig. 5a). In loosely packed samples this phenomenon was only observed in permafrost samples, but not in active layer samples. As expected, active layer samples showed exponential $CO_2$ accumulation only when stirred, likely because stirring increases substrate availability. Together, this might suggest that microbial growth is a constitute part of post-thaw resuscitation response in permafrost peat. Fitting the accumulation curves to a mixed growth model produced realistic specific growth rates (Eq. 1). It has to be noted, however, that the

estimated specific growth rates after thawing had no repercussions for long-term decomposition; plotting cumulative $CO_2$ production after 350 days over μ did not reveal any significant relationship. Largest decomposition was observed with initial μ values ~ 0.03 (data not shown).

Constantly stirred slurries were included as treatment to explore the effect of relieving diffusional constraints for $O_2$ and substrates during aerobic and anaerobic metabolism. As expected, constant stirring of slurries increased

$CO_2$ production and $O_2$ consumption in the first phase of the incubation. However, the addition of water to the slurries decreased the headspace volume which meant that the absolute amount of $O_2$ present in the oxic slurries was smaller than in the oxic incubations with loose peat, which biases the comparison between loosely paced and slurried oxic incubation (Fig. S4 to S9). In general, differences in cumulative $CO_2$ production between loosely packed samples and slurries were small (Fig. S3).

The finding that $CO_2$ production of TZ and PF samples was comparable to that of active layer samples suggests that C quality does not limit microbial decomposition of permafrost peat after thawing, despite having been frozen for centuries (Fig. 6). This was true for all three sites despite marked variations in peat decomposability among the sites. The common depth pattern of degradability could be related to similar formation history of the three studied peat plateaus (Table 1). The top of the active layer, where the highest rates were measured, includes the

root zone with continuous input of fresh plant litter (Fig. 6). This was also reflected in high C/N and $\delta^{13}C$ values in this layer at all three sites (Fig. 4). In the lower parts of the active layer, the peat has likely been exposed to aerobic conditions for decades or even centuries, without input of fresh plant litter. It may therefore be strongly decomposed and thus depleted of labile C which would explain the lower observed degradation rates for AL2 and AL3 samples (Fig. 6). On the other hand, the PF layers contain frozen peat produced under wetland (i.e. mostly

anaerobic) conditions which has not been exposed to decomposition yet. Here, more labile C may be available which could explain the secondary peak in degradation rates observed for TZ and PF1 samples (Fig. 6). Another explanation to why C degradation varied over depth at Áidejávri and Lakselv could be the increasing content of iron over depth (Fig. 4) which can trap organic C and limit mobilisation and degradation (Patzner et al., 2020). However, this cannot explain C degradation at Iškoras since this site overall has very little iron (Fig. 4).

Differences in C decomposition potentials between the sites might also be explained by difference in formation history or site-specific environmental factors. A study in a Swedish peat plateau found that reductive dissolution of iron during permafrost thaw can lead to increase in $CH_4$ emissions (Patzner et al., 2022). This might explain why Lakselv and Áidejávri, both having high iron contents, showed a faster increase in $CH_4$ production compared to Iškoras (Fig. 4 and 7).

**4.2 Total C mobilisation**

Initially oxic samples from all sites did not produce $CH_4$ for 350 days even after turning anoxic, indicating that methanogenesis was inhibited beyond the depletion of $O_2$ (Fig. S4, S6, S8). Yet, it cannot be ruled out that longer incubation could have resulted in $CH_4$ production (Knoblauch et al., 2018). DOC was consumed to a larger degree

in initially oxic than anoxic PF samples (Fig. 8). This may be partly due to less DOC degradation under anoxic conditions, or it could be related to the higher extractability of DOC at higher pH in anoxic samples (Table S10). In incubations of thawed cores from a Finnish peat plateau, Panneer Selvam et al. (2017) found that DOC from the active layer had a lower initial degradation potential than DOC from thawed permafrost. This agrees with our finding that $CO_2$ accumulation in permafrost samples was faster than in deeper active layer samples despite releasing less DOC, suggesting that 'old' permafrost DOC was more degradable than 'new' DOC in the active layer (Fig. 6 and 8). This trend was especially evident at Áidejávri where $CO_2$ accumulation in the top permafrost exceeded that of the top active layer, despite releasing more DOC in the active layer. Still, the substantial release of DOC from the active layer is of concern because DOC is easily lost along water paths and can potentially increase the $CO_2$ production downstream (Panneer Selvam et al., 2017; Voigt et al., 2019). The DOC run-off from permafrost affected areas has been found to both increase C emission from fresh waters as well as contributing a significant amount of C export to the Arctic Ocean (Vorobyev et al., 2021).

Differences in gas kinetics for both $CO_2$ and $CH_4$ among the three peat plateaus (Fig. 5) could be related to differences in abundance and taxonomic composition of microbial communities. We tried to work aseptically during field sampling and laboratory sample handling, but cross-contamination (e.g. by the corer) cannot be ruled out. Notwithstanding, our results suggest that permafrost samples harbour competent microbial taxa which proliferate over time as seen by the exponential product accumulation in some of our incubations. An incubation study with permafrost soil from the Tibetan Plateau found that $CO_2$ release after thawing was positively related to functional gene abundance for C degradation (Chen et al., 2020) without involving changes of the taxonomic composition. In general, high-altitude ecosystems differ from high-latitude ecosystems by having lower contents of OM and ice, with consequences for microbial community composition (Wang and Xue, 2021). In the present study, there were clear differences in the kinetics of gaseous product accumulation, both for $CO_2$ and $CH_4$ (Fig. 5 and Table 2). Thawed permafrost peat from Iškoras supported exponential $CO_2$ accumulation and PF peat from Áidejávri exponential $CH_4$ accumulation, strongly indicating microbial growth which will eventually result in community change. Detailed molecular studies would be needed to elucidate whether post-thaw community change leads to overall more degradation of permafrost C, or to a shift in the $CO_2/CH_4$ ratio under anoxic conditions.

## 4.3 Thermokarst peat decomposition

Inundation in thermokarst may be the ultimate fate of permafrost peat from thawing peat plateaus. It involves mixing with unfrozen peat, extended anoxia when inundated and buried by sediments and access to fresh C from autotrophic production in the ponds. Our results showed that the potential $CO_2$ production of permafrost peat thawed in situ in a thermokarst pond (TK-PF1) was smaller than of permafrost peat thawed ex situ (PF), but still in the same order of magnitude (Table S3). On the other hand, $CH_4$ production potentials over 96 days were several orders of magnitude higher in thermokarst samples than in undisturbed PF samples (Table S4). This may be due to proliferation of a highly productive methanogenetic community over time, but also due to additional nutrient input from surrounding non-PF fens or bogs (In 'T Zandt et al., 2020).

The observed increase in $CH_4$ production between 96 and 350 days of incubation for permafrost samples (Fig. 7), especially those from Lakselv and Áidejávri, suggests that proliferation of a functioning methanogenetic community after permafrost thaw takes time and depends, among others, on the fate of permafrost peat (e.g.

inundation) after peat plateau collapse. This could mean that $CH_4$ production measured ex situ greatly underestimates the true $CH_4$ production potential of in situ thawed material, even in longer-term incubations. A similar conclusion can be drawn from the findings of Knoblauch et al. (2013) who incubated permafrost material from Holocene and Late Pleistocene permafrost sediments of the Lena River Delta in Siberia and found that $CH_4$ production reached maximum rates after an average of 2.6 years of incubation.

In our study, most $CH_4$ was produced by thermokarst peat from Iškoras (Table S4); this could be related to the fact that permafrost thaw occurred more recently, with labile C still being present. The Áidejávri thermokarst site, on the other hand, has been thawed for a longer time which might explain why the C is more stable resulting in less methanogenesis. Another explanation might be differences in environmental factors such as soil moisture, temperature and vegetation composition (Olefeldt et al., 2013).

Similar to our study, Kirkwood et al. (2021) incubated both peat plateau and thermokarst peat anoxically and found higher $CH_4$ emissions in the thermokarst compared to active layer and permafrost samples after 225 days (Table S7). High pH was found to be a good predictor for potential $CH_4$ production in the Canadian thermokarst. In our study, the opposite was the case; $CH_4$ production was highest in Iškoras thermokarst with lower pH than in Áidejávri (Table S4 and S5), suggesting that local differences in peat quality and time since thawing play a role for the $CH_4$ emission potential.

**5 Conclusion**

This study evaluates the C degradability from active layer, transition zone and permafrost at three peat plateau sites in northern Norway through ex situ incubations. In addition, samples from thermokarst adjacent to the peat plateaus were investigated at two of the sites. Observed C degradation rates varied among the three sites, while all three sites showed similar degradation patterns over depth with largest $CO_2$ production in the top of active layer and a second maximum in permafrost layers. High-resolution post-thaw gas kinetics showed marked differences in microbial growth response which, however, did not affect long term C mineralisation potentials. The main limitation for C degradation was $O_2$ availability. Significant $CH_4$ production was only observed in samples from the transition zone and permafrost layers after prolonged anoxic incubation. $CH_4$ production increased over time showing that methanogenesis could play an important role in C degradation under prolonged anoxic conditions. This was further supported by thermokarst samples that showed two to four orders of magnitude larger $CH_4$ production rates as compared to freshly thawed peat plateau samples. DOC released during incubation of active layer peat plateau samples exceeded gaseous C release. Our incubation study indicates that burial of permafrost peat in thermokarst and DOC runoff to down-stream ecosystems should be taken account for when estimating C degradation in collapsing peat plateau ecosystems.

**Data availability**

Data are available on Zenodo https://doi.org/10.5281/zenodo.10696561.

**Competing interests**

The authors declare that they have no conflict of interest.

**Acknowledgements**

We would like to thank the laboratory staff at the Faculty of Environmental Sciences and Natural Resource Management, NMBU, especially Trygve Fredriksen, Pia Frostad and Solfrid Lohne.

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
