# Peer review of "Carbon degradation and mobilisation potentials of thawing permafrost peatlands in Northern Norway"

_EGUsphere, 2024_

## Author Comment (AC1)

Written in black are comments from the reviewers.

Written in blue are comments from the authors (modifications made in the text are in italic).

All line numbers refer to the R1 version of this manuscript.

**RC1: 'Comment on egusphere-2024-562', Anonymous Referee #1, 26 Mar 2024**

**Summary:**

Kjær et al. presents an incubation study using soils from permafrost peatlands in northern Norway. The authors collected material from active to mineral layer in a permafrost habitat, and a comparison core from thawed thermokarst soil. The authors used aerobic and anaerobic headspaces in the incubations, at different depths from an inland-to-ocean gradient of permafrost peatland habitats. The first 20 days were sampled multiple times a day and give insights to the microbial activity directly after permafrost material thaws. Each depth group was divided in four with two vials being kept at field moisture, and the other two vials having water added for a slurry mixture. The replicates were further divided so that one vial begins in an aerobic headspace and one vial remains anaerobic for the duration of the incubation. The authors provide rates and cumulative values for the O2, CO2, N2, CH4 and N2O and geochemical data for the soil cores. Stable isotope signature analysis and dissolved organic carbon samples were also taken to supplement the gas production dataset. The main findings of the study were that the three sites had similar patterns by depth in the various parameters measured. Notably, the active layer samples were highly productive in DOC production, a pattern not observed in the post-thaw samples.

I am enthusiastic about the content of the study, and especially of the high-resolution gas measurements. The paper is pertinent and could be of interest to the scientific community with its high-resolution gas measurements, and strategic selection of interesting study sites. However, multiple major comments should be considered.

We thank the reviewer for the positive reception of our study and address major and minor comments below.

**MAJOR COMMENTS:**

The authors clearly state their objectives with the paper, though a hypothesis was not provided. This was reflected further in the paper when the results were more reports of the data and did not prove / disprove what the authors had hypothesized how the various depths would respond to the treatment groups.

R1.1 We did not have a hypothesis when we started the experiment as it was entirely unclear what results we would get. At the same time depth-resolved measurements on peat permafrost cores are extremely rare. Our goal was therefore to explore which parameters are site specific and which general across the landscape of permafrost peatlands. We have added the following sentence to the Introduction: *As depth-resolved measurements are extremely rare in permafrost peat, the goal was to explore which parameters are site specific and which are general across a permafrost peatland* (L. 67).

Some measurements (ie. stable isotope signatures) are reported in the results but not further integrated into the rest of the paper.

R1.2 We report stable isotope signatures of the peat core samples as a part of the general geochemical characterisation rather than overinterpreting week isotopic signals. Below is a graph from my master thesis which plots $\delta^{15}$N over C/N values from three peat cores. According to Conen et al. (2013) and Krüger et al. (2017), this plot can be used to indicate the state of perturbation in different permafrost layers. We did not see a clear pattern in our samples and have therefore excluded this story from the manuscript. We also refer to Claire Treat's community comment below: "Not everything in the thesis needs to be included in the paper as the thesis counts as a reference…". The master thesis can be downloaded under https://hdl.handle.net/11250/2788732.

[Figure]

Figure 45: Correlation between $\delta^{15}$N values and C/N ratio across depth profiles at the three peat plateaus Iškoras, Áidejávri and Lakselv. A: The solid line shows the relationship between $\delta^{15}$N values and C/N ratio given by Conen et al. (2013) to distinguish unperturbed and perturbed samples. Unperturbed samples are shown by closed markers and are within the uncertainty envelope ±2.4‰ $\delta^{15}$N of unperturbed samples (solid and dashed lines). Perturbed samples are outside the dashed line (±2.4‰ $\delta^{15}$N). As described in Krüger et al. (2017), samples above the uncertainty envelope indicate increase of mineralisation due to permafrost uplift (dashed markers) and samples below cryoturbation (hollow markers). B: Classification from A shown as depth profile

Further analysis could also be done with the cumulative gas production between coring sites. The novelty of this study lies with its high-resolution measurements, especially the gas production from the first ~20 days; the community could benefit from the quantification of this fast carbon pool that is largely not captured with ex-situ incubations to date.

R1.3 Monitoring the immediate metabolic response to thawing of peat permafrost at high resolution was indeed one of our objectives. We compared the gas kinetics (not the cumulative production) to those of unfrozen peat and tentatively assigned exponential product accumulation to microbial growth in the original manuscript. We have now extended this approach by quantitatively evaluating growth kinetics from high-resolution measurements during the first 17 to 19 days of incubation. This was done by fitting a growth equation (Eq. 1) combining a common growth model with constant rate for non-growing decomposers to measured $CO_2$ accumulation following the framework of Stenström et al. (1998). The approach is described in lines 220-229. We also decided to move table S3, reporting the apparent specific growth rates to the main text (now table 2) to highlight the findings from the period of high-resolution gas monitoring. We also modified the Introduction to include immediate post-thaw response as a research question (L. 56-69). The results are now discussed it in more detail in lines 262-272 (Results) and lines 413-422 (Discussion). At the same time, we removed table S4 showing exponentiality of $CH_4$ accumulation as most samples had long lag phases with exponential $CH_4$ accumulation occurring far beyond high-resolution gas monitoring.

I am not entirely convinced that the paper in its current form presents a robust statistical analysis to meet the authors objectives. The largest issue being that there were many treatment groups but no technical replicates. A replicate of n=1 is statistically weak, though the high resolution of the measurements provides some compensation for this. Overall, I think that the site/depth replicates could be binned differently to explore stronger hypotheses that are developed in response to the comment above. T-test's were performed to determine difference between the geochemical parameters of the sites but further analysis could be done on the gas fluxes than determining if the production rate is exponential or not.

R1.4 The reviewer is correct that we do not have replicates, reflecting the explorative nature of our study. The reasons for this decision are predominately logistically: retrieving complete permafrost peat cores including active layer, transition zone, permafrost and a mineral bottom

layer and bringing undisturbed samples to the laboratory entails significant logistic efforts. Also, the number of samples which could be incubated simultaneously in the incubator was limited (~40 bottles + standards). Instead of replicating peat plateau samples at a single site, we chose to spread the effort and sampled one complete profile at each of the three distinct sites, which were chosen to span the different climatic conditions under which peat plateaus exist in Norway (Borge et al., 2017). With respect to technical replication in the laboratory, we abstained from dividing samples into subsamples (except for the four treatments oxic/loose, anoxic/loose, oxic/slurry and anoxic/slurry) as we were mostly interested in vertical trends rather than statistical differences between functional layers. We agree with the reviewer that samples from the same functional layer could be binned for statistical analyses (pseudo-replication), but we do not see any statistical need for doing so for the time being. We now justify our approach in line 190: *For logistical reasons (number of bottles which could be incubated simultaneously) and a limited amount of sample material per depth, we were not able to replicate samples for each of the treatments. We therefore choose to focus on different incubation treatments rather than replicates per depth and explored differences along the vertical profiles at different sites under different conditions.* To further explore the high-resolution gas data, as proposed by the reviewer, we have reevaluated $CO_2$ kinetics (Eq. 1; L. 220-229). See reply R1.3 for more details.

Important parameters such as the general vegetation of field sites, soil type, and soil bulk density were not reported. These parameters are essential for the comparison of these results to similar experiments, and usability for those looking to use the data for reviews or inputs for earth system models.

R1.5 Unfortunately, we did not have the resources to carry out a detailed vegetation analyses or to determine bulk densities; instead, we now refer to other studies from the same sites in lines 89-95; *The dominant vegetation at the peat plateaus has been characterised as dwarf shrubs, mosses, lichen and cloudberry herbs at Iškoras and Lakselv (Kjellman et al., 2018; Martin et al., 2019). Thermokarst and surrounding wet fen areas are dominated by sedges, cotton grass and Sphagnum sp. (Kjellman et al., 2018; Martin et al., 2019). Áidejávri has similar vegetation, however, this site has not been investigated and described in detail. The soils at the three peat plateaus are characterised as histosols (IUSS Working Group WRB 2014). Typical bulk densities in peat plateaus of the region range from 0.08 and 0.28 g cm-3 in the organic layers (Kjellman et al., 2018; Mats Rouven Ippach, pers. com.).*

The authors reference the cores being collected on an inland-to-sea gradient. This was mentioned in the introduction and site description but not further discussed. The natural gradient a transect introduces a range of variables (ie. climate, vegetation, etc.) that likely influence the data and could give greater relevance to the patterns seen in this study. It would be of value for the authors to shape their discussion of what research questions they were looking to answer with this gradient when they selected their soil coring locations.

R1.6 The reviewer is right, and we have therefore changed the Abstract (L. 14) and Introduction (L. 53) to clarify that we did not work with a strict gradient but rather a selection of sites varying in climate, altitude, peat and permafrost age. As mentioned before, we choose this approach to explore a larger area of discontinuous permafrost. The terms 'inland-to-sea gradient' and 'chronosequence' have been removed from the text.

**MINOR COMMENTS:**

Line 34: Some grammatical errors, some rephrasing for clarity needed.

We have rephrased the sentence.

Line 43: Here I would suggest expanding on what has been introduced here. Anaerobic respiration in peatlands is just starting to be understood, and this experiment's high resolution measurements afford a unique opportunity to capture processes that happen right as the material starts to thaw from its frozen state.

We added two sentences to the Introduction: *Environmental factors controlling the degradation of organic matter from thawing peat plateaus are still poorly understood* (L. 44) and *We used both short- and long-term incubations at moderate temperature (10°C); short-term incubations to explore the immediate metabolic response with and without oxygen, as little is known about the resuscitation kinetics after controlled thawing of permafrost peat.* (L. 56).

Line 45: Although it could be considered "common knowledge", I think it's worth briefly elaborating on why CO2, CH4 and carbon cycles are important on a global scale in the introduction.

We added a sentence in line 36: *Both $CO_2$ and $CH_4$ are greenhouse gases that contribute to global warming, and increased C release from Arctic regions may thus amplify the global warming (Knoblauch et al., 2018).*

Line 50: This is the first mention of the sea / inland dynamic. A paragraph on the variables this natural gradient introduces would improve the paper and better emphasize the importance of the data. Clearly stating which sea or body of water is referenced would make for a more easily readable paper.

*We have removed the term sea-to-inland gradient. The text now reads (L. 52): In this study, we determined post-thaw degradation kinetics of peat from three Norwegian permafrost peatlands. The peat plateaus were selected to represent an area with sporadic permafrost differing in peat and permafrost age, as well as in climatic conditions.*

Line 51-52: A brief discussion of why active layer / permafrost layer vary in oxygen availability and potential for degradation would improve paper. Suggested readings, citations.

*We thank the reviewer for this suggestion and the readings. We now elaborate on the role of oxygen for peat decomposition in permafrost peatland in the Introduction (L. 56): We used both short- and long-term incubations at moderate temperature (10°C); short-term incubations to explore the immediate metabolic response with and without oxygen, as little is known about the resuscitation kinetics after controlled thawing of permafrost peat; long-term incubation to evaluate differences in degradability of active layer and permafrost peat which have been shown to differ greatly with $O_2$ availability and temperature (Kirkwood et al., 2021; Treat et al., 2014; Waldrop et al., 2021; Panneer Selvam et al., 2017). Permafrost peat differs distinctly in $O_2$ availability between the active layer and the permafrost, with the former being drier and more exposed to $O_2$ than the latter. However, $O_2$ availability in the active layer varies greatly during the year since its frozen in winter and thaws over the summer (Åkerman and Johansson, 2008). The collapse of peat plateaus and sequential formation of thermokarst ponds also changes the $O_2$ availability and thus the degradation potentials (Hodgkins et al., 2014).*

Clymo, I doR. S., & Hayward, P. M. (1982). The Ecology of Sphagnum. In A. J. E. Smith (Ed.), Bryophyte Ecology (pp. 229–289). Springer Netherlands. https://doi.org/10.1007/978-94-009-5891-3_8

Hodgkins, S. B., Tfaily, M. M., McCalley, C. K., Logan, T. A., Crill, P. M., Saleska, S. R., Rich, V. I., & Chanton, J. P. (2014). Changes in peat chemistry associated with permafrost thaw increase greenhouse gas production. Proceedings of the National Academy of Sciences, 111(16), 5819–5824. https://doi.org/10.1073/pnas.1314641111

Schädel, C., Beem-Miller, J., Aziz Rad, M., Crow, S. E., Hicks Pries, C. E., Ernakovich, J., Hoyt, A. M., Plante, A., Stoner, S., Treat, C. C., & Sierra, C. A. (2020). Decomposability of soil organic matter over time: The Soil Incubation Database (SIDb, version 1.0) and guidance for incubation procedures. Earth System Science Data, 1511–1524. https://doi.org/10.5194/essd-12-1511-2020

Line 56: A definition and introduction of thermokarst earlier in the introduction would improve the flow and better emphasize the importance of the data

We have added a sentence (L. 45): *Thermokarst ponds are the natural succession to peat plateaus after thawing and are thus crucial for understanding future permafrost degradation.*

Line 60: Consider rephrasing for clarity

We have rephrased the sentence to make it clearer (L. 72): *Additionally, to explore the role of microbial growth in C degradation and mobilisation, a parallel set of samples were incubated as stirred soil slurries for 96 days, thus eliminating diffusional constraints on substrate availability.*

Line 80: While the site history on a geologic scale is informative, a description of the modern vegetation at the time of sampling, soil types at each depth, and a broader site introduction is necessary.

In addition to the geologic scale and soil type, the site description now also includes information about vegetation and bulk densities (see reply to R1.5).

Line 104: How were the layers decided? Visual inspection of the soil horizons, or was there a metric used to delineate a change in the soil core? Example photos of vegetation or soil of sites would be useful in the supplement, if possible.

We divided AL samples upon visual inspection in the field which is now detailed in L. 120: *The active layer samples were assigned upon a visual inspection. AL1 and AL3 were sampled from the top and bottom of the active layer, respectively, and AL2 from the middle. AL1 did not contain surface vegetation but was less decomposed than AL2 and AL3.* The photo below shows the sampling of a frozen core. It was difficult to obtain a good photo of the active layer since we did not want to disturb the site to much and kept the hole small.

The frozen core was sampled in 5 cm increments and immediately transferred to centrifuge tubes. Therefore, it is not possible to show a picture of the entire frozen core. All frozen samples

were shipped to the laboratory, and assigned to different functional layers, a selection of which was used for incubation and chemical analyses. This is now described as (L. 123): *The TZ sample was taken from the top of the frozen core and the PF1 sample just below this. PF3 was taken from the bottom of the frozen core which consisted of mineral soil at Áidejávri and Lakselv. PF2 samples were taken in between PF1 and PF3.*

[Figure]

Line 179: A brief description of how the authors went from the GC output to their reported units of umol CO2/CH4 g-1 dw-1 and µg g dw-1 225 d-1 using the cited Molstad et al method would increase the reproducibility of the values used in this study and readability of paper.

We added a short description in line 208: *After converting peak areas to ppmv, moles of CO₂ and CH₄ accumulated or O₂ consumed were calculated taking account of dissolution in peat*

*water (Wilhelm et al., 1977; Appelo and Postma, 1993), dilution by He back-pumping and leakage of $O_2$ during sample admission. For more details, see Molstad et al. (2007).*

Line 190: For the statistics, it is unclear if the peat characteristics applies to the geochemical analysis or the gas chromatography measurements. If the t-test, etc was performed in a statistics program (ie. R, python, etc) it should be stated and the package credited.

We used Excel for Microsoft 365 for the geochemical comparison. This is now noted in line 229. For estimating specific growth rates, we used Sigmaplot 14 which is now indicated in line 228.

Line 216: Whether or not the gas production is exponential / linear at different timepoints has some value, but the larger question is if the rate of production is different between the treatment groups (slurried or field moisture, aerobic or anaerobic, the inland-to-sea gradient between field sites, and depth groups). Here the study's largest weakness of having no replicates becomes an issue. Its difficult to make assertions about these treatments inducing higher or lower rates of GHG production with one technical replicate per treatment

In response to the reviewer's fair criticism, we introduced a growth model (see reply R1.3) to quantitatively estimate apparent specific growth rates. We obtained specific growth rates ranging from 0.007 to 0.07 $h^{-1}$ which is in the range of reasonable generation times (14 -145 h) for incubation experiments. Based on these values we now discuss differences between sites, depths and treatments. The following text has been added in the Discussion (L. 413): *Incubating the samples as stirred slurries versus loosely placed peat affected initial gas kinetics. We observed exponential $CO_2$ accumulation within the first 20 days of incubation in some of the bottles, indicating microbial growth (Table 2, Fig. 5a). In loosely packed samples this phenomenon was only observed in permafrost samples, but not in AL samples. As expected, AL samples showed exponential $CO_2$ accumulation only when stirred, likely because stirring increases substrate availability. Together, this might suggest that microbial growth is a constitute part of post-thaw resuscitation response in permafrost peat. Fitting the accumulation curves to a mixed growth model produced realistic specific growth rates (Eq. 1). It has to be noted, however, that the estimated specific growth rates after thawing had no repercussions for long-term decomposition; plotting cumulative $CO_2$ production after 350 days over μ did not reveal any significant relationship. Largest decomposition was observed with initial μ ~ 0.03 (data not shown).*

Line 254: Were the incubations "constantly agitated"? The methods state that the slurry treatment groups were mixed for an hour to disperse the peat, then the peat settled

We thank the reviewer for noticing this mistake. The slurries were not constantly stirred after the initial period, and we have removed "constantly" from the sentence.

Line 257: Samples completely under the water line are mostly in an anaerobic environment. I would suspect that the inhibited rate of O2 diffusing into the water would be the limitation, and not the reduced headspace volume.

We agree that the lack of stirring would inhibit $O_2$ diffusion. However, we observed rapid $O_2$ take down in several stirred slurries resulting in $O_2$ limitation much earlier than in the corresponding loosey placed samples. This has to be attributed to the smaller initial $O_2$ amount in the slurried bottles (which had a smaller headspace) than the packed bottles. The kinetics for both loosely packed samples and slurries are now shown in Fig. S4 to S9.

Line 265: Table could be moved to Supplement, and values of interest simply discussed in the results.

We moved table 2 to the Supplementary Information (now called table S3).

Line 275: It is likely not that the thawing permafrost was necessarily "stimulated" but rather that the authors are observing a lag phase common in ex-situ incubation studies with permafrost soils.

Knoblauch, C., Beer, C., Liebner, S., Grigoriev, M. N., & Pfeiffer, E.-M. (2018). Methane production as key to the greenhouse gas budget of thawing permafrost. Nature Climate Change (4), 309–312. https://doi.org/10.1038/s41558-018-0095-z

We thank the reviewer for this remark. The sentence has now been changed to (L. 329): *Yet, while $CO_2$ production slowed down over time, $CH_4$ production increased (Fig. 5b) indicating that methanogenesis in thawing PF peat had a lag phase before producing $CH_4$ (Knoblauch et al., 2018).*

Line 279: Same comment as Line 254

Same answer as Line 254.

Line 305: Same comment as Line 265

We moved table 3 to the Supplementary Information (now called S4).

Line 314-317: Needs Figure references and more specific language

We added references to the figures and rephrased the text.

Line 324-328: This paragraph would benefit from references to Figures and/or specific references to the data.

We added references to figures.

Line 339: The authors imply that the CO2 production from their study is higher than that of another permafrost site using a similar methodology. While this is useful, I would recommend a reassessment of this interpretation. Incubations are useful in their ability to isolate individual ecosystem-scale controls on GHG production; however, they are limited in direct comparisons of the production values as (the authors state in Line 350) there are many discrepancies between incubation studies.

We agree and have added a disclaimer warranting caution when comparing rates across different studies (L. 381): *Although caution is warranted when comparing decomposition rates across different incubation studies, Kirkwood et al. (2021) found, similarly to our study, highest CO$_2$ release in the top of the active layer for Canadian peat plateaus (see Supplement in Kirkwood et al. (2021)) and for some sites a second maximum in the permafrost peat.*

Line 373: Bringing in the stable isotope results and expanding on what the significance of the different patterns of depletion / enrichment in each site and by depth here would add value to the discussion

See R1.2.

Line 381: I suggest moving Figure S4 into the main body of the manuscript and discussing this dataset more. I would also suggest expanding on how these were analyzed in the methods.

We now include the data of figure S4 in figure 4 and we have expanded on the methods. None of these elements had a significant effect on decomposition rates. Therefore, we do not further discuss these data.

Line 393: For this study (Panneer Salvem et al 2017) and the others used in this discussion, some explanation is needed as to why the authors chose these particular studies to compare their results so closely to the comparison papers results.

There are few studies with these organic-rich soils in sporadic permafrost areas. The references were chosen because they deal with peat plateaus similar to our sites.

Line 405: There has been significant focus on exponential / non-exponential production rates, but no reason given as to why this is significant to answering a hypothesis or adds value to the study.

Exponential product accumulation is a strong indicator of microbial growth (Stenström et al., 1998; 2001) and as such of interest in a study on peat permafrost thawing. See also R1.3.

Line 408: It is unclear why this high-altitude / high-latitude comparison is being made

There are not many studies looking at post-thaw microbial succession. Chen et al. (2020) and Wang and Xue (2921) studied soils from the Tibetan plateau, a high-altitude ecosystem, which will differ from high latitude systems in several respects.

Line 419-420: Same comment as Line 324-328

We added references to figures.

Line 443: For the conclusion of this paper, the authors should consider framing the study more robustly into the significance of these high-resolution measurements into the current state of knowledge on permafrost C dynamics.

We now added a sentence concluding on the observed high-resolution kinetics (L. 514): *High-resolution post-thaw gas kinetics showed marked differences in microbial growth response which, however, did not affect long term C mineralisation potentials.*

Line 453: Perhaps an earlier mention of these elements (DOC runoff and down-stream ecosystems) if it is the main recommendation of the study. For example, expanding on the statement in Line 400 would achieve this.

We added a sentence about the significance of DOC runoff (L. 463). *The DOC run-off from permafrost affected areas has been found to be both increasing C emission from fresh waters as well as contributing a significant amount of C export to the Arctic Ocean (Vorobyev et al., 2021).*

References:

Molstad, L., Dörsch, P., and Bakken, L. R.: Robotized incubation system for monitoring gases (O2, NO, N2O N2) in denitrifying cultures, Journal of microbiological methods, 71, 202-211, https://doi.org/10.1016/j.mimet.2007.08.011, 2007.

Clymo, R. S. and Hayward, P. M.: The Ecology of Sphagnum, Smith, A.J.E. (eds) Bryophyte Ecology, Springer, Dordrecht, https://doi.org/10.1007/978-94-009-5891-3_8, 1982.

Conen, F., Yakutin, M. V., Carle, N. & Alewell, C. (2013). δ15N natural abundance may directly disclose perturbed soil when related to C:N ratio. Rapid Commun Mass Spectrom, 27 (10): 1101-1014. doi: 10.1002/rcm.6552.

Hodgkins, S. B., Tfaily, M. M., McCalley, C. K., Logan, T. A., Crill, P. M., Saleska, S. R., Rich, V. I., and Chanton, J. P.: Changes in peat chemistry associated with permafrost thaw increase greenhouse gas production, Proceedings of the National Academy of Sciences, 111(16), 5819–5824, https://doi.org/10.1073/pnas.1314641111, 2014.

Knoblauch, C., Beer, C., Liebner, S., Grigoriev, M. N., and Pfeiffer, E.: Methane production as key to the greenhouse gas budget of thawing permafrost, Nature Clim Change, 8, 309–312, https://doi.org/10.1038/s41558-018-0095-z, 2018.

Krüger, J. P., Conen, F., Leifeld, J. & Alewell, C. (2017). Palsa Uplift Identified by Stable Isotope Depth Profiles and Relation of δ15N to C/N Ratio. Permafrost and Periglacial Processes, 28 (2): 485-492. doi: 10.1002/ppp.1936.

Martin, L. C. P., Nitzbon, J., Aas, K. S., Etzelmüller, B., Kristiansen, H., and Westermann, S.: Stability conditions of peat plateaus and palsas in northern Norway, Journal of Geophysical Research: Earth Surface, 124, 705–719, https://doi.org/10.1029/2018JF004945, 2019.

Stenström, J., Stenberg, B., and Johansson, M.: Kinetics of Substrate-Induced Respiration (SIR): Theory, Ambio, 27(1), 35–39. http://www.jstor.org/stable/4314682, 1998.

Stenström, J., Svensson, K., and Johansson, M.: Reversible transition between active and dormant microbial states in soil, FEMS Microbiology Ecology, 36, 2-3, 93–104, https://doi.org/10.1111/j.1574-6941.2001.tb00829.x, 2001.

Vorobyev, S. N., Karlsson, J., Kolesnichenko, Y. Y., Korets, M. A., and Pokrovsky, O. S.: Fluvial carbon dioxide emission from the Lena River basin during the spring flood, Biogeosciences, 18, 4919–4936, https://doi.org/10.5194/bg-18-4919-2021, 2021.

Wilhelm, E., Battino, R. and Wilcock, R. J.: Low-pressure solubility of gases in liquid water, Chemical Reviews, 77 (2), 219-262, https://doi.org/10.1021/cr60306a003, 1977.

Åkerman, H. J. and Johansson, M.: Thawing permafrost and thicker active layers in sub-arctic Sweden, Permafrost Periglac. Process., 19, 279-292, https://doi.org/10.1002/ppp.626, 2008.

**RC2: 'Comment on egusphere-2024-562', Anonymous Referee #2, 12 Apr 2024**

This manuscript presents interesting findings regarding carbon degradation in permafrost peat cores, stratified by depth and sampled from various regions. The key findings indicate comparable CO2 losses between the active layer and permafrost/transition zone, with DOC production surpassing gaseous C losses, and CH4 emissions occurring following inundation of thawed peat material in thermokarst ponds. These results offer significant and new insights into carbon dynamics in permafrost environments.

We thank the reviewer for the positive reception of our study and address comments below.

The manuscript is well-written, logically structured, and engaging to read. One issue is is the lack of technical replication which have been thoroughly addressed already by other reviewers.

R2.1 We acknowledge the lack of technical replication as a weakness of our study and refer to the response given to reviewer #1 (R1.4). However, we think that the high depth resolution provides some compensation for the lack of replicates.

Another one is the presentation of the data as gram per incubation duration. In this manner, the data cannot be readily compared to other studies or generalized. It would be advantageous to present the data so that they can be used fo modeling carbon dynamics in different permafrost soils. Therefore, I recommend reporting the data on a gram (of mol) of dry soil or soil carbon basis (e.g., mg CO2-C g-1 d-1 or mg CO2-C g-1 C-1).

R2.2 We chose to report cumulative rates for two incubation periods, i.e. $0 – 96d$ after which slurry incubations were discontinued, and $0 – 350d$ covering the entire incubation period. These numbers can easily be divided by time to arrive at hourly or daily rates. We further chose to keep the units as they are. Carbon content, organic matter content and rates are given in figure 4 and SI (https://doi.org/10.5281/zenodo.10696561), to calculate rates per gram C.

Ideally, decay rates should be computed by fitting them into a first-order models with one or more pools, as it has been practiced by others. Please look at the paper by Schädel et al. (2020) Earth Sys. Sci. Data, 12, on how to report data from incubation studies to include them into a larger database on decomposability of SOC.

R2.3 In principle, we agree with the reviewer that our data could be used along the lines of Schädel et al. (2020) for fitting pool models. However, we also note that our product accumulation kinetics indicated microbial growth, particularly so in thawed permafrost

samples (inserts in Fig 5a), which questions the validity of fitting $CO_2$ accumulation to steady-state pool models. We have uploaded our data into a database (https://doi.org/10.5281/zenodo.10696561) which can be used for modelling.

Regarding specific comments, Line 264 starting with "CO2 production in thermokarst cores....". is not supported by the data (referencing Table 2), and later the discussion highlights decreasing CO2 fluxes in thermokarst cores, likely attributed to water inundation.

R2.4 We changed the sentence in the Results section to (L. 315) *CO₂ production in thermokarst cores (Iškoras and Áidejávri) was in the same order of magnitude (40-241 μmol CO₂ g dw⁻¹ 96 days-1) as the permafrost cores but CO₂ production from new peat at Áidejávri was somewhat higher (616 μmol CO₂ g dw⁻¹ 96 days⁻¹) (Table S3) and in the Discussion Section to (L. 484): Our results showed that the potential CO₂ production of permafrost peat thawed in situ in a thermokarst pond (TK-PF1) was smaller than of permafrost peat thawed ex situ (PF), but still in the same order of magnitude (Table S3).*

By the way, can you also provide soil moisture data from the cores, to substantiate that conclusion? It is important to get information about future CO2 fluxes in these ecosystems, and the MS makes a significant contribution here.

R2.5 We now provide soil moisture data from the cores in Table S11.

In Figure 8, it would be beneficial to clearly indicate that the bars are stacked to facilitate comparison between the two components (CO2/DOC). I would maybe even present them next to each other, to make the comparison easier.

R2.6 We agree that it is difficult to see that they are stacked. We have therefore prepared a coloured graph. We also tried to place them next to each other, but the comparison was even more difficult then.

Finally, while the data and discussion on dissolved organic carbon (DOC) and lateral runoff are really interesting, the significance of extracting DOC before and after one year of incubation warrants consideration. I am wondering what a DOC extract before and after one year of incubation really tells us. DOC is permanently produced and consumed, the net change must be very small as compared to the gross changes. Which makes it even more difficult to believe that cumulative CO2 emissions are lower than net DOC production in these soils, since DOC is very labile and thus heavily mineralized. Can it be that this enrichment in DOC is due

to die-off of microbes after such a long incubation? I encourage the authors to double-check their calculations and report the proportion of carbon lost as CO2/CH4-C from the initial carbon pool after 350 days, along with the proportion that would have been produced as DOC and is thus susceptible to leaching (in percentage), together with some critical evaluation of the DOC data presented.

R2.7 We agree that gross rates of DOC production and consumption over a year may be large with only small changes in net DOC. Yet, it is this net change, which was positive in AL samples which determines the partitioning between gaseous and dissolved C losses and hence the long-term fate of post-thaw released C. It is possible that some of the released DOC is related to microbial death going along with decreasing substrate availability, but this is besides the question. Microbial populations adapt to the carrying capacity of a soil by adjusting their growth and death rates, the net effect being DOC net consumption or production. The DOC in the closed bottles likely became recalcitrant with time while C was recycled. This would explain why we find smaller net DOC production in oxic than anoxic samples (Figure 8) as more gaseous C was produced and removed from the DOC pool. It is interesting to note that larger DOC immobilization in Iškoras PF than Áidejávri and Lakselv samples (Figure 8) went along with higher DOC concentrations upon thawing (Figure 4), indicating a substantial sink for post-thaw released DOC at this site. We have double-checked our calculations.

**CC1: 'Comment on egusphere-2024-562', Claire C. Treat, 05 Apr 2024**

Kjaer et al. use a high depth-resolution of core sections from three separate palsas in northern Fennoscandia to look at potential C losses with permafrost thaw. They measure peat properties, potential CO2 and CH4 production under aerobic and anoxic conditions as well as the DOC released during the incubation. The strengths and unique features of this experiment include the high resolution incubation data, the combination of both gaseous and dissolved pathways of C loss, and the inclusion of a thawed replicate for comparison and the use of three individual sites. Additionally, the paper is relatively easy to read although the discussion section needs reference to the figures in the main text. Most importantly, the results support the conclusions There have been several incubation experiments previously of permafrost peats; these are appropriately referred to in the study and the discussion.

We thank Claire Treat for her constructive criticism. We have rewritten the Discussion section and included reference to tables and figures.

What I'm missing is what this study adds to the earlier ones. I'm also missing what from this study can be generalized because the interpretations of the results are very specific to each site and also quite qualitative. The conclusions section redeems this but this is the only place where the findings are presented so clearly. A revision of the paper, particularly the results and discussion, keeping in mind the key points from the conclusion is warranted in order to streamline. In the overall justification of the study, the explanation (hypotheses) of why the differences in C production (CO2, CH4, DOC) might occur is missing, beyond stating that post-thaw degradation kinetics might depend on peat quality and formation history (line 49).

R3.1 We have worked thoroughly through the Results and Discussion sections and sharpened the message where possible. We now emphasise the high-resolution post-thaw gas kinetics, which is novel, and also added a new figure and some more discussion of the $DOC:CO_2$ partitioning. Our study is and remains a descriptive study with no specific initial hypothesis. Yet, we cover three distinct permafrost peat cores in one geographical region, which supposedly differ in peat histories and chemistry (see below). Unlike other authors, who pooled samples from different functional layers, we present high-resolution depth profiles of potential C degradability and DOC production, which allowed us to compare the post-thaw degradation potential of permafrost peat with that of active layer peat. We find that degradation potentials of PF peat almost equalled those of AL peat, despite being 'old carbon'. We highlight this finding in the Conclusion section (L. 512).

I think there is an earlier study that might help with the interpretation here. This study is using a chronosequence of timing of permafrost formation as well as variable timing of peatland initiation: Treat et al. (2021) explores the influence of the "residence time in the active layer" on the potential carbon losses using modeling. I think that would be a really useful theoretical framework to adopt here for the development of the hypotheses, data analysis, and interpretation of the results if the goal is to link peat quality and formation history with potential C production, which I think is appropriate and interesting.

R3.2 We thank Claire Treat for proposing a theoretical framework for our data. The problem is that we are lacking peat-chronological data for one of the sites (Áidejávri), which would be needed to estimate the residence time of peat in the active layer. Moreover, there may be many other factors determining peat chemistry and recalcitrance to microbial decomposition: Apart from peat and permafrost age, the three sites also differed in topographic setting (e.g. altitude, distance from the sea). A new study is on its way, specifically designed to explore the nexus between peat/permafrost/thermokarst chronology and degradability at two of the sites (Iškoras and Áidejávri). These new data will also be used for modelling. It is therefore beyond the scope of the present study to apply the proposed modelling framework. We now mention this in the Discussion section (L. 402): *A standardised incubation protocol would improve the comparability among studies and help validating and improving numerical models of C dynamics in permafrost peatlands (e.g. Treat et al., 2021) which are critical tools to quantify present and future climate change impacts on these sensitive ecosystems.*

There are a few further challenges in this study, both with the experimental design and with the interpretation. The main issue with the experimental design is the lack of replicates, at least in the current analysis. The analysis uses different (categorical) depths and use different treatments, one replicate of each treatment per depth. The replicates are the sites. However, sites aren't used as replicates; instead, they are discussed individually but always quantitatively. The main problem that I have with this approach is that it really limits the interpretation and application of these results beyond these specific study site (as well as not allowing for statistics). The authors could explore different methods of binning which could increase the number of replicates and might make the trends clearer and potentially allow statistical analyses. Some more creative analyses may help in linking some of the various measures, such as PCAs or NMDS.

R3.3 All three reviewers commented on this, and we agree that more replicates would have been an advantage if we were looking for statistical trends over depth or between functionally equivalent layers at different sites. In our descriptive study, given the logistic constraints (see R1.4) we chose to focus on depth profiles at geographically distinct sites rather than proving statistically significant differences between sites or depths. Multivariate statistical analyses can be found in the MSc thesis (https://hdl.handle.net/11250/2788732) in which this manuscript has its source.

**Additional comments**

Table 1 could include info on timing of peat formation and permafrost formation, particularly since it's used as a justification for the study design.

We added information about timing of peat and permafrost formation to Table 1.

Tables 2 and 3 are not very intuitive. I think this point is important, is there a more effective way to get this across? Maybe as additional lines in Figure 6 and 7?

We agree and have moved the tables to the Supplementary Information (table S3 and S4). The numbers are now referred to directly in the text.

Not everything in the thesis needs to be included in the paper as the thesis counts as a reference, for example the slurry vs loose peat experiment does not show up in the main conclusion and could be referenced in the methods section.

We agree and we use the MSc thesis actively in this rebuttal. A reference to the thesis is now also given in the SI. We also added $O_2$ consumption kinetics of stirred slurries to the SI (Figures S5, S7 and S9). Differences between loose and slurry incubations are now addressed in more detail in connection with estimating specific growth rates from initial $CO_2$ accumulation (L. 268-272 and L. 413-422)

This study does a nice job summarizing earlier aerobic:anaerobic CO2 produciton ratios in section 4.1; perhaps a table might be a nice way to summarize this.

We give a range of aerobic to anaerobic $CO_2$ production ratios in the text (L. 406) We have also added a more detailed table to SI (Table S12) showing anoxic $CO_2$ production as percentage of oxic $CO_2$ production throughout 350 days.

389-390: I think it's quite unlikely that oxic samples would produce methane as long as the headspace remained oxic.

Some initially oxic samples turned anoxic during the incubation period (See fig. S4 to S9). We did not observe any CH$_4$ production during the initially oxic phase. We now write (L. 451): *Initially oxic samples from all sites did not produce CH$_4$ for 350 days even after turning anoxic, indicating that methanogenesis was inhibited beyond the depletion of O$_2$ (Fig. S4, S6, S8).*

Conclusions: currently doesn't discuss the aerobic:anaerobic production ratios that are a major point in the discussion

We have added a sentence to the Conclusion: *The main limitation for C degradation was O$_2$ availability* (L. 515).

**From EGU:**

Notification to the authors:

I just noticed that your figures 2 and 3 contains aerials. If you are not the originator of the images, then appropriate credit or copyright must be given.'

We thank the editors for noticing this. We have updated the figure text with credits and copyright information.

---

## Author Response (AR2)

Written in black are comments from the reviewers and editor.

Written in blue are comments from the authors (modifications made in the text are in italic).

All line numbers refer to the R2 version of this manuscript.

**Associate editor decision: Reconsider after major revisions**

Dear authors,

I send back to review your manuscript to one of the previous reviewer and the way you deal with replication is still an issue. Your study is interesting and deserve publication but you need to find a way around this problem.

Best

Bertrand Guenet

Thank you for your feedback and for the encouragement to continue with the publication of our study. In response to the reviewer's comment on technical replicates, we have modified the title and changed the Abstract and Conclusion sections to better reflect the findings that can be drawn from our approach. We have in particular replaced formulations that imply a broader generalization of our results, and instead rather refer to the actual results of our study.

**RC1: Report #1, Submitted on 12 Jun 2024, Anonymous referee #1**

The authors revisions answered the input of the reviews, though two important open questions remain

1) The lack of technical replicates remains an issue for this paper. The authors addressed this by reframing parts of the paper, especially with the edits at line 210. However, the new text and the paper as a whole is framed more as a pilot study of permafrost soil depth-dependency and less of of comparison between different habitats. The text (especially the title, abstract and conclusion) should be updated to reflect this.

We have changed the abstract and conclusions to better reflect the actual findings of our study rather than generalizing them, given the lack of technical replication. We have also changed the title to highlight the methodological constraints of the study: *Carbon degradation and mobilisation potentials of thawing permafrost peatlands in Northern Norway inferred from laboratory incubations.*

2) In line 105 of the revised document, it is mentioned that the bulk density of the soil was estimated in the active layer but (perhaps I missed it in the text, supplemental info) remains unreported for the permafrost layer. Other important parameters such as soil moisture remain unreported. Especially for the high depth resolution, these parameters would likely have a wide range of values and impact on the gas flux measurements, especially in the Transition Zone depth.

We thank the reviewer for spotting this issue which was indeed unclear. We have revised the text to clearly state that the values reported in Kjellman et al. (2018) refer to both active layer and permafrost peat (line 95). Furthermore, we now also provide a reference to the water contents of our samples which are reported in Table S11 (line 98).